# Eve3D: Elevating Vision Models for Enhanced 3D Surface Reconstruction via Gaussian Splatting

Jiawei Zhang[1][*]    Youmin Zhang[2][†]    Fabio Tosi[4]    Meiying Gu[1]    Jiahe Li[1]    Xiaohan Yu[5]
Jin Zheng[1,3]    Xiao Bai[1][‡]    Matteo Poggi[4]

[1]School of Computer Science and Engineering,  State Key Laboratory of Complex Critical
Software Environment,  Jiangxi Research Institute,  Beihang University
[2]Rawmantic AI    [3]State Key Laboratory of Virtual Reality Technology and Systems, Beijing
[4]University of Bologna    [5]Macquarie University

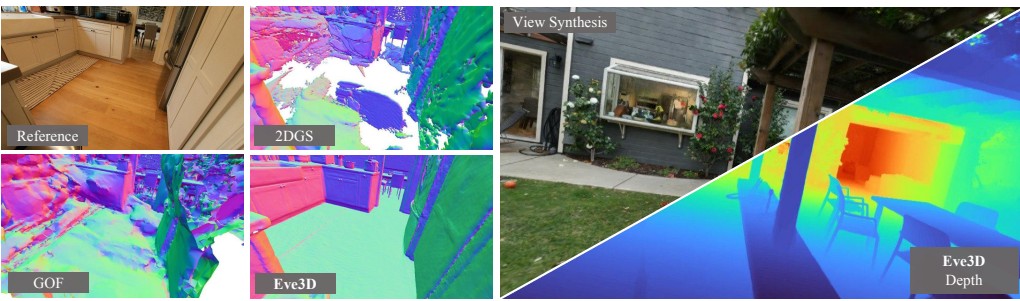

Figure 1: **Geometric Reconstruction and Rendering.** Examples from a challenging scene [11] showing our Eve3D method's accurate geometry and appearance. Left: Normal maps highlight superior surface reconstruction over 2DGS [18] and GOF [55], particularly on flat surfaces and object boundaries. Right: Photorealistic view synthesis alongside its detailed depth map rendering.

## Abstract

We present Eve3D, a novel framework for dense 3D reconstruction based on 3D Gaussian Splatting (3DGS). While most existing methods rely on imperfect priors derived from pre-trained vision models, Eve3D fully leverages these priors by jointly optimizing both them and the 3DGS backbone. This joint optimization creates a mutually reinforcing cycle: the priors enhance the quality of 3DGS, which in turn refines the priors, further improving the reconstruction. Additionally, Eve3D introduces a novel optimization step based on bundle adjustment, overcoming the limitations of the highly local supervision in standard 3DGS pipelines. Eve3D achieves state-of-the-art results in surface reconstruction and novel view synthesis on the Tanks & Temples, DTU, and Mip-NeRF360 datasets. while retaining fast convergence, highlighting an unprecedented trade-off between accuracy and speed.

## 1   Introduction

Dense 3D scene reconstruction is a crucial task in computer vision and graphics, supporting applications ranging from virtual reality and simulating environments to robotic navigation. Recent breakthroughs in this field draw inspiration from the adjacent literature concerning novel view

---

[*]This work was conducted during Jiawei Zhang's research internship at Rawmantic AI.

[†]Project Lead

[‡]Corresponding author

39th Conference on Neural Information Processing Systems (NeurIPS 2025).

synthesis [32, 22], which faced a remarkable revolution in the last few years. Initially, Neural Radiance Fields (NeRFs) [32] were adapted to encode dense 3D surfaces within MLP weights [42, 25]; however, despite their compact representation, these approaches impose prohibitive computational demands, often requiring hundreds of hours of processing to reconstruct a single scene. More recently, 3D Gaussian Splatting [22] (3DGS) has gained significant attention in the graphics community, characterized by an optimized rasterizer that facilitates real-time rendering, establishing it as the current preferred alternative to NeRFs. This has led to the development of numerous 3DGS-based approaches [18, 55, 6, 29] that deliver promising results and high-fidelity scene reconstructions. However, despite the efficiency and flexibility of 3DGS and the latest advances, the framework continues to face significant limitations that hinder its broader application for 3D reconstruction task.

First and foremost, 3DGS in its original formulation is unsuited for this purpose, as Gaussian primitives rarely fit accurately to true surface geometries [16]. This limitation derives from two critical factors: the inherent independence of Gaussian primitives, which operate without contextual awareness, and the sole reliance on image reconstruction losses during training. While this approach excels at novel view synthesis, it often fails at properly modeling the real 3D geometry of the observed scene. Second, the accuracy of 3DGS heavily relies on a proper initialization strategy for seeding Gaussian primitives. Despite its efficiency and widespread adoption, COLMAP's [35] reliance on local feature makes it vulnerable to challenging scenarios such as textureless regions and repetitive patterns, resulting in structural distortions, blurring, and under-reconstructed areas.

To address these shortcomings, recent 3D reconstruction pipelines built upon 3DGS [18, 55, 6, 29] have incorporated additional scene *priors* from pre-trained vision models [48, 58, 4, 5, 12, 1, 15, 53]. Among these supplementary signals, *depth* and *surface normals* have proven most effective, significantly enhancing reconstruction quality in traditionally challenging scenarios such as textureless regions. On the one hand, although vision models can reason both locally and globally over their inputs, they are constrained by the limited amount of images they can process simultaneously – e.g., typically a single frame [48, 12, 1, 15, 53], a stereo pair [58, 44, 3], or at most about ten images [4, 5]. This limitation restricts these models to analyzing only localized portions of a scene when extracting priors. Furthermore, despite their training on large-scale datasets ranging from hundreds of thousands [4, 5] to several million images [48], the priors they predict still suffer from inaccuracies. These dual constraints create sub-optimal supervision for 3DGS, tampering with the optimization process and ultimately limiting the final 3D reconstruction accuracy.

In this paper, we introduce **Eve3D**, a novel 3D reconstruction framework built upon 3D Gaussian Splatting, specifically designed to overcome these limitations. Firstly, Eve3D unlocks the full potential of vision models by rendering synthetic rectified stereo pairs through 3DGS and applying state-of-the-art stereo foundation models to obtain geometric priors. These priors are then refined through joint optimization alongside the 3DGS model itself. This joint optimization strategy works by back-propagating gradients through both the depth rendered by 3DGS and the predicted depth priors, with the latter treated as learnable parameters throughout the process. Secondly, we propose a local bundle adjustment strategy that maintains global consistency across co-visible frames during each forward optimization. This approach overcomes 3DGS's inherent limitation in simultaneously rasterizing and optimizing multiple frames – a constraint imposed by computational complexity. This allows Eve3D to achieve unprecedented accuracy, as demonstrated in Fig. 1.

Eve3D is trained and evaluated over popular benchmarks for 3D reconstruction, including DTU [9] and Tanks and Temples [24], achieving state-of-the-art accuracy on both datasets. In particular, on the latter, this is done in just 20 minutes, while it takes only $\sim 1$ GPU hours to push the accuracy to the upper bound. In summary, the main contributions of this paper are:

- We introduce Eve3D, a novel framework for dense 3D reconstruction based on 3DGS, setting a new state-of-the-art in the field while maintaining efficient training time.

- We develop an improved supervision paradigm for 3DGS used to train Eve3D, which treats external priors as learnable parameters and optimizes them jointly with the 3DGS model during training.

- We introduce a local bundle adjustment to better enforce multi-view consistency during each optimization step, overcoming one of the main limitations of 3DGS-based frameworks.

## 2 Related Work

**Neural Scene Representations.** Neural scene representations have revolutionized 3D reconstruction and rendering. NeRF [32] pioneered this direction with continuous volumetric functions modeled by MLPs, though at high computational cost. More recently, 3DGS [22] introduced an explicit representation using 3D Gaussian primitives with efficient rasterization-based rendering, achieving superior quality and real-time performance. Despite successful applications in large-scale scene reconstruction [23, 28, 26], SLAM systems [47, 21, 37, 63, 40], dynamic scene modeling [49, 50, 46, 27], AI-generated content [38, 8, 34, 61], and autonomous driving [62, 30, 60], both NeRF and 3DGS focus primarily on appearance rather than geometry, resulting in poorly defined surfaces.

**Neural Surface Reconstruction.** To address these limitations, several methods have extended neural rendering for accurate surface reconstruction. NeuS [42] and VolSDF [52] represent surfaces as zero-level sets of signed distance functions without mask supervision. Neuralangelo [25] enhances detail with multi-resolution hash grids and numerical gradients, while NeuralWarp [10] improves consistency through patch warping. These implicit approaches produce high-quality surfaces but require significant computational resources and long optimization times.

**Surface Reconstruction with Gaussian Splatting**. 3DGS's efficiency has inspired many surface reconstruction methods. SuGaR [16] introduced regularization for surface-aligned Gaussians, enabling Poisson reconstruction. 2DGS [18] uses planar disks for surface modeling, while Gaussian Surfels [9] treats local z-axis as normal direction. GS2Mesh [45] extracts meshes using TSDF fusion on depth maps from a pre-trained stereo model, while StereoGS [33] employs self-improving depth supervision with virtual stereo pairs. Some approaches combine Gaussians with implicit fields: GOF [55] derives an opacity field, while GSDF [54] and 3DGSR [31] integrate 3DGS with signed distance functions. Others focus on geometric constraints, like PGSR [6] with unbiased depth rendering, DN-Splatter [41] with depth/normal priors, and VCR-GauS [7] with view-consistent depth-normal regularization. GS-Pull [57] aligns Gaussians to a neural SDF's zero-level set. Despite progress, current approaches have limitations with geometric priors - either using external vision model priors that may be inconsistent [48, 58, 4, 5, 12, 1, 15, 53], or applying heuristic constraints that struggle with complex geometries. Our work jointly optimizes Gaussians and priors in a unified framework, introducing non-local information flow for better geometric consistency.

## 3 Method

### 3.1 Framework Overview

**Scene Representation.** As illustrated in Fig. 2, our backbone is built over 3D Gaussian Splatting, which models the scene as a set of Gaussian primitives, each one defined as:

$$\mathcal{G}_i(\boldsymbol{x}|\boldsymbol{\mu}_i, \boldsymbol{\Sigma_i}) = e^{-\frac{1}{2}(\boldsymbol{x}-\boldsymbol{\mu}_i)^\top \boldsymbol{\Sigma}_i^{-1}(\boldsymbol{x}-\boldsymbol{\mu}_i)} \tag{1}$$

with $\boldsymbol{\mu}_i \in \mathbb{R}^3$ and $\boldsymbol{\Sigma}_i \in \mathbb{R}^{3\times3}$ being the center and 3D covariance matrix, respectively. The latter can be decomposed into scaling and rotation matrices $\boldsymbol{S}_i\,\boldsymbol{R}_i \in \mathbb{R}^{3\times3}$:

$$\boldsymbol{\Sigma}_i = \boldsymbol{R}_i\boldsymbol{S}_i\boldsymbol{S}_i^\top\boldsymbol{R}_i^\top \tag{2}$$

To better fit surfaces [18, 9], we enforce Gaussians to be flat by minimizing the minimal factor in $\boldsymbol{S}_i$. The pixel-wise color $\boldsymbol{C} \in \mathbb{R}^3$ is rendered through $\alpha$-blending:

$$\boldsymbol{C} = \sum_{i \in N} T_i\alpha_i\boldsymbol{c}_i, \quad T_i = \prod_{j=1}^{i-1}(1-\alpha_i), \tag{3}$$

where $\alpha$ and $\boldsymbol{c}_i \in \mathbb{R}^3$ are the alpha and view-dependent color. Similarly, properties such as depth and surface normals can be rendered. Following [6], normals $\boldsymbol{N}$ are derived from the minimum scale factor direction $\boldsymbol{n}_i$ and camera rotation $\boldsymbol{R}_c$. Depth $\boldsymbol{D}$ are obtained through unbiased rendering [6], calculated as the intersection between the ray and the plane defined by rendered normals and the distance map $\mathcal{D}$, where $d_i = (\boldsymbol{R}_c^T(\boldsymbol{\mu}_i - \boldsymbol{T}_c))^T(\boldsymbol{R}_c^T\boldsymbol{n}_i)$ represents the distance from the plane to the camera center $\boldsymbol{T}_c$. The two geometric properties follow:

$$\boldsymbol{N} = \sum_{i \in N} \boldsymbol{R}_c^T\boldsymbol{n}_i\alpha_i\prod_{j=1}^{i-1}(1-\alpha_j), \quad \mathcal{D} = \sum_{i \in N} d_i\alpha_i\prod_{j=1}^{i-1}(1-\alpha_j). \tag{4}$$

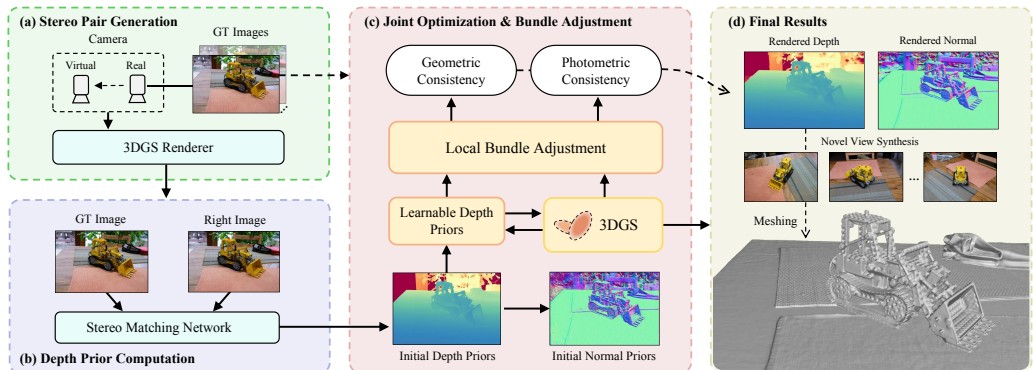

Figure 2: **Overview of our Framework.** (a) We generate stereo pairs by rendering from a virtual camera using 3DGS. (b) A pre-trained stereo network extracts initial depth priors. (c) Our core contribution: joint optimization of learnable depth priors and 3DGS, enhanced by local bundle adjustment for geometric and photometric consistency. (d) Final results showing high-quality mesh reconstruction, accurate depth/normal maps, and high-fidelity novel view synthesis.

While color can be supervised with real images, depth and normals can be supervised using *priors* from pre-trained vision models [48, 58, 4, 5, 12, 1, 15, 53] for better optimization.

**Priors Computation.** Unlike approaches supervising 3DGS with ill-posed monocular depth models [48, 12], we use binocular stereo matching [39] for more geometrically consistent priors. Since our method operates in a standard multi-view setting with a single moving camera rather than a stereo rig, we exploit 3DGS's view synthesis to render rectified stereo pairs [33, 45]. Specifically, for a given camera pose $\mathbf{P}_i$, we define a virtual right camera with pose $\mathbf{P}_i^r$, at a distance $b$ as:

$$\mathbf{P}_i^r = \begin{pmatrix} \mathbb{I} & \mathbf{t} \\ 0 & 1 \end{pmatrix} \times \mathbf{P}_i \qquad \text{with} \quad \mathbf{t} = \begin{pmatrix} b & 0 & 0 \end{pmatrix}^\top \tag{5}$$

We render a virtual right image and form a stereo pair with the input left image $(\mathbf{I}_i, \mathbf{I}_i^r)$. We then apply a pre-trained state-of-the-art stereo model – FoundationStereo [44] in our implementation – to predict a disparity map, which is converted to depth $D^*$ through triangulation. We also derive a confidence mask $M^c$ by computing the consistency check between disparity maps for left and right views. While effective for assisting 3DGS optimization [33] and recovering meshes [45], we argue that explicitly addressing the noisy nature of these priors is critical to fully exploit their potential.

## 3.2 Prior-Involved Local Bundle Adjustment

Multi-view consistency is vital for determining accurate surfaces. However, extracting accurate and detailed surfaces from either Gaussians or vision models presents challenges without multi-view constraints. This difficulty arises because 3DGS captures depth information through image rendering supervision, while vision models inherently contain noise and have limited view inputs. To address this issue, we have developed a local bundle adjustment algorithm that utilizes both rendered and prior depth maps, enhancing their multi-view consistency to represent surfaces more accurately. In multi-view pairs, we employ priors at source views, which allows us to incorporate more viewpoints in a single loop by eliminating the need for depth rendering. For the reference view, we select either the rendered or prior depth map, depending on the stage of joint optimization (Sec 3.3).

For view $V_i$ at the current iteration, we build a factor-graph $(\mathcal{V}, \mathcal{E})$ to perform local bundle adjustment. To balance reconstruction quality and efficiency, rather than optimizing all frames together, we only select views that overlap with the current view $i$. Specifically, following standard MVS methods [51, 4, 5], we determine neighboring views by computing an overlapping score and select only the top $K$ frames $\{V_j\}_{j=1}^K$. We construct the graph by adding edge connections between the current view $V_i$ and each of its neighboring views $V_j$. The underlying optimization principle enforces both geometric and photometric consistency.

**Geometric Consistency.** Given learnable depth $\hat{D}_i$ of the current view $V_i$, we first convert it into a normal map $\mathcal{N}_{\hat{D}_i}$ with finite differences as in [18], and derive the distance map $\hat{\mathcal{D}}_i$ as:

$$\hat{\mathcal{D}}_i(\boldsymbol{p}) = \hat{D}_i(\boldsymbol{p})\mathcal{N}_{\hat{D}_i}^T(\boldsymbol{p})K_i^{-1}\hat{\boldsymbol{p}}, \tag{6}$$

where $\boldsymbol{p}$ is the 2D position on the image plane, $\hat{\boldsymbol{p}}$ denotes its homogeneous coordinate, and $K_i$ is the camera intrinsic matrix. We then map the full set of pixel coordinates $P_i$ from the current view $V_i$ to the neighboring view $V_j$ through the homography matrix $H_{ij}$:

$$\hat{P}_j = H_{ij}P_i, \quad H_{ij} = K_j(R_{ij} - \frac{T_{ij}\mathcal{N}_{\hat{D}_i}^T}{\hat{\mathcal{D}}_i})K_i^{-1} \tag{7}$$

where $R_{ij}$ and $T_{ij}$ are the relative rotation and translation from view $V_i$ to the neighboring view $V_j$. Similarly, for pixels in the neighboring view $V_j$, we derive the surface normal and distance map from its learnable depth $\hat{D}_j$ to compute the homography matrix $H_{ji}$. By enforcing geometry consistency, we aim at minimizing the projection error $\min \Phi_{ij}$, where

$$\Phi_{ij} = \| P_i - H_{ji}H_{ij}P_i \| . \tag{8}$$

**Photometric Consistency**. This constraint is based on plane patches. For pixels $P_i$ in the current view $V_i$, we map a $7 \times 7$ pixel patch centered at each pixel $\boldsymbol{p} \in P_i$ to the neighboring view using the homography matrix $H_{ij}$. We aim to minimize the photometric error to zero:

$$\Psi_{ij} = (1 - \text{NCC}(I_i(P_i), I_j(H_{ij}P_i)), \tag{9}$$

**Objective.** To account for occlusions between views and noise in depth estimates, following [6], we model the confidence map to weight the error function $W_{ij}$ and the overall cost function to refine depth maps of the current view and its neighbors is defined as:

$$\mathcal{L}_{lba} = \lambda_{lba} \sum_{(s,t)\in\mathcal{E}} W_{st}(\lambda_g\Phi_{st} + \lambda_p\Psi_{st}). \tag{10}$$

### 3.3 Joint Optimization of 3DGS and Priors

Regardless of their source, all model-generated priors inevitably contain noise and inaccuracies. Rather than treating them as rigid supervision, we jointly optimize both the 3DGS model and the associated guidance. This creates a mutually beneficial relationship, refining priors through multi-view consistency while providing improved supervision for the 3DGS representation.

**Parameterized Prior.** Given a prior depth map $D^*$, we initialize a set of learnable parameters $\hat{D}$ with its values. The joint optimization follows a two-phase schedule. Initially (i.e., before iteration $T_{joint}$), we use the original $D^*$ to supervise 3DGS and avoid early convergence to poor local minima. Later, once the 3DGS-rendered depth becomes sufficiently reliable, we switch to optimizing $\hat{D}$ jointly with 3DGS parameters through backpropagation, allowing the model to refine both the scene representation and the the guidance signal simultaneously without degrading their quality.

**Local Bundle Adjustment Pre-training.** Neural networks that generate prior depth estimates often do so at very sparse viewpoints, resulting in insufficient multi-view consistency. To address this limitation, before starting the joint optimization ($T_{joint}$), we apply our proposed local bundle adjustment between the prior depths of different viewpoints, which will provide a multi-view consistent prior initialization before starting the joint optimization of 3DGS and prior depth.

**Confidence Mask Update.** As the quality of depth prior gradually improves through the joint optimization and local bundle adjustment, the initial confidence mask $M^c$ may become outdated. Concurrently, the confidence map computed during local bundle adjustment reflects the quality of the latest depth prior. Therefore, we update the mask $\hat{M}^c$ as:

$$\hat{M}_i^c = \hat{M}_i^c \vee (W_{ij} > 0) \ \text{ for } j \in \{1, 2, \cdots, K\} \tag{11}$$

For pixels with value 0 in $\hat{M}^c$, i.e., low-quality depth prior, it remains possibile for them to be adjusted to a more accurate position by 3DGS:

$$\mathcal{L}_{pull} = \lambda_{pull}(\sim \hat{M}^c)\|\hat{D} - D^{\text{detach}}\|_1, \tag{12}$$

where $D^{\text{detach}}$ denotes the gradient of rendered depth $D$ is detached. When such pixels attain a sufficiently accurate state and are subsequently classified as confident by local bundle adjustment, they become eligible to participate in joint optimization and local bundle adjustment, ultimately contributing to accurate surface reconstruction.

Table 1: **Quantitative results of F1 Score on Tanks and Temples.**   ,   ,    indicate the absolute, second, and third bests respectively.

| | Method | Barn | Caterpillar | Courthouse | Ignatius | Meetingroom | Truck | Mean ↑ | Time |
|---|---|---|---|---|---|---|---|---|---|
| Implicit | NeuS [42] | 0.29 | 0.29 | 0.17 | 0.83 | 0.24 | 0.45 | 0.38 | >24h |
| | Geo-Neus [14] | 0.33 | 0.26 | 0.12 | 0.72 | 0.20 | 0.45 | 0.35 | >24h |
| | Neuralangelo [25] | 0.70 | 0.36 | 0.28 | 0.89 | 0.32 | 0.48 | 0.50 | >128h |
| Explicit | 3DGS [22] | 0.13 | 0.08 | 0.09 | 0.04 | 0.01 | 0.19 | 0.09 | 20m |
| | SuGaR [16] | 0.14 | 0.16 | 0.08 | 0.33 | 0.15 | 0.26 | 0.19 | 2h |
| | DN-Splatter [41] | 0.15 | 0.11 | 0.07 | 0.18 | 0.01 | 0.20 | 0.12 | 1h |
| | GSurfels [9] | 0.24 | 0.22 | 0.07 | 0.39 | 0.12 | 0.24 | 0.21 | 15m |
| | 2DGS [18] | 0.36 | 0.23 | 0.13 | 0.44 | 0.16 | 0.26 | 0.30 | 34m |
| | GOF [55] | 0.51 | 0.41 | 0.28 | 0.68 | 0.28 | 0.58 | 0.46 | 2h |
| | PGSR [6] | 0.66 | 0.44 | 0.20 | 0.81 | 0.33 | 0.66 | 0.52 | 45m |
| | GS-Pull [57] | 0.60 | 0.37 | 0.16 | 0.71 | 0.22 | 0.52 | 0.43 | 38m |
| | *Eve3D-fast (Ours)* | 0.69 | 0.44 | 0.34 | 0.82 | 0.41 | 0.62 | 0.56 | 20m |
| | **Eve3D (Ours)** | 0.70 | 0.48 | 0.35 | 0.83 | 0.46 | 0.66 | 0.58 | 1.2h |

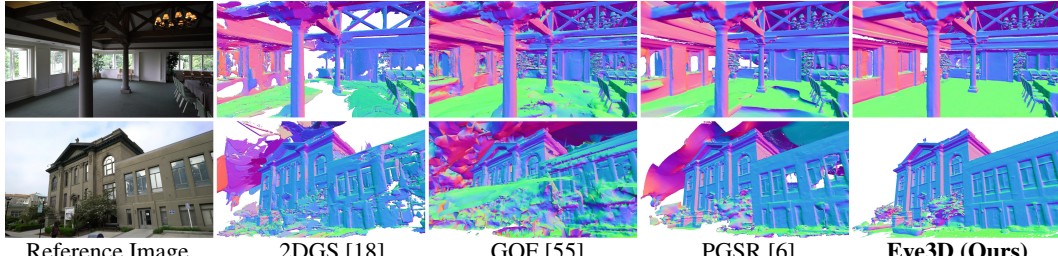

| Reference Image | 2DGS [18] | GOF [55] | PGSR [6] | **Eve3D (Ours)** |

Figure 3: **Qualitative Comparison on Tanks and Temples.** We visualize the surface normal of reconstructed 3D meshes for comparison.

### 3.4 Training Loss

With the prior depth, we apply a single-view prior loss $\mathcal{L}_{depth}$ to regularize the depth and normal from 3D Gaussians in confident regions weighted by $\hat{M}^c$:

$$\mathcal{L}_{prior}^{depth} = \hat{M}^c \odot \begin{cases} ||D^* - D||_1, & \text{iter} < \text{T}_{\text{joint}}, \\ ||\hat{D} - D||_1, & \text{otherwise.} \end{cases} \tag{13}$$

To strengthen the impact on plane surface reconstruction, we also jointly optimize the normals obtained from 3DGS and the depth prior in confident regions as follows:

$$\mathcal{L}_{prior}^{normal} = \hat{M}^c \odot \begin{cases} \Gamma(\mathcal{N}_{D^*}, \mathcal{N}_D) + \Gamma(\mathcal{N}_{D^*}, N), & \text{iter} < \text{T}_{\text{joint}}, \\ \Gamma(\mathcal{N}_{\hat{D}}, \mathcal{N}_D) + \Gamma(\mathcal{N}_{\hat{D}}, N), & \text{otherwise,} \end{cases} \tag{14}$$

where $\Gamma(A, B)$ denotes the pixel-wise cosine distance $(1 - A \cdot B)$ between normal maps. In addition, we also use ground-truth color loss $\mathcal{L}_c$ [22], depth-normal consistency loss $\mathcal{L}_{dn}$ to encourage consistency between rendered depth and rotational normal [6, 18, 55], and scale loss $\mathcal{L}_s$ [6] to encourage Gaussians to flatten to planes. The final training loss is:

$$\mathcal{L} = \mathcal{L}_c + \mathcal{L}_{dn} + \mathcal{L}_s + \mathcal{L}_{prior} + \mathcal{L}_{lba} + \mathcal{L}_{pull}. \tag{15}$$

## 4 Experiments

### 4.1 Experimental Setup

**Datasets.** We evaluate on real-world datasets, including object-centric, indoor, and outdoor scenes. For 3D reconstruction, we use large-scale scenes from Tanks and Temples [24] and 15 object-centric scenes from DTU [20]. For novel view synthesis, we use the Mip-NeRF360 dataset [2].

**Evaluation Metrics.** Following established protocols, we assess reconstruction accuracy using Chamfer Distance (CD) on DTU [9] and F-score on Tanks and Temples [24], employing the official evaluation scripts. For novel view synthesis evaluation on Mip-NeRF360, we use standard rendering quality metrics: PSNR, SSIM [43], and LPIPS [56].

**Baselines.** We compare against state-of-the-art methods from two categories: (1) implicit NeRF-based approaches including NeRF [32], VolSDF [52], NeuS [42], Geo-Neus [14], NeuralWarp [10], and Neuralangelo [25]; and (2) explicit 3DGS-based frameworks including 3DGS [22], SuGaR [16], DN-Splatter [41], GSurfels [9], 2DGS [18], GOF [55], GS2Mesh [45], PGSR [6], and GS-Pull [57].

Table 2: **Quantitative Results (Chamfer Distance) on DTU.** ▓, ▓, ▓ indicate the absolute, second, and third bests respectively.

| | Method | 24 | 37 | 40 | 55 | 63 | 65 | 69 | 83 | 97 | 105 | 106 | 110 | 114 | 118 | 122 | Mean↓ | Time |
|---|---|---|---|---|---|---|---|---|---|---|---|---|---|---|---|---|---|---|
| Implicit | NeRF [32] | 1.90 | 1.60 | 1.85 | 0.58 | 2.28 | 1.27 | 1.47 | 1.67 | 2.05 | 1.07 | 0.88 | 2.53 | 1.06 | 1.15 | 0.96 | 1.49 | > 12h |
| | VolSDF [52] | 1.14 | 1.26 | 0.81 | 0.49 | 1.25 | 0.70 | 0.72 | 1.29 | 1.18 | 0.70 | 0.66 | 1.08 | 0.42 | 0.61 | 0.55 | 0.86 | >12h |
| | NeuS [42] | 1.00 | 1.37 | 0.93 | 0.43 | 1.10 | 0.65 | 0.57 | 1.48 | 1.09 | 0.83 | 0.52 | 1.20 | 0.35 | 0.49 | 0.54 | 0.84 | >12h |
| | NeuralWarp [10] | 0.49 | 0.71 | 0.38 | 0.38 | 0.79 | 0.81 | 0.82 | 1.20 | 1.06 | 0.68 | 0.66 | 0.74 | 0.41 | 0.63 | 0.51 | 0.68 | >10h |
| | Neuralangelo [25] | 0.37 | 0.72 | 0.35 | 0.35 | 0.87 | 0.54 | 0.53 | 1.29 | 0.97 | 0.73 | 0.47 | 0.74 | 0.32 | 0.41 | 0.43 | d0.61 | >12h |
| Explicit | 3DGS [22] | 2.14 | 1.53 | 2.08 | 1.68 | 3.49 | 2.21 | 1.43 | 2.07 | 2.22 | 1.75 | 1.79 | 2.55 | 1.53 | 1.52 | 1.50 | 1.96 | 12m |
| | SuGaR [16] | 1.47 | 1.33 | 1.13 | 0.61 | 2.25 | 1.71 | 1.15 | 1.63 | 1.62 | 1.07 | 0.79 | 2.45 | 0.98 | 0.88 | 0.79 | 1.33 | 1h |
| | DN-Splatter [41] | 1.60 | 2.03 | 1.42 | 1.44 | 2.37 | 2.11 | 1.62 | 1.95 | 1.88 | 1.48 | 1.63 | 1.82 | 1.20 | 1.50 | 1.40 | 1.70 | 30m |
| | GSurfels [9] | 0.66 | 0.93 | 0.54 | 0.41 | 1.06 | 1.14 | 0.85 | 1.29 | 1.53 | 0.79 | 0.82 | 1.58 | 0.45 | 0.66 | 0.53 | 0.88 | 11m |
| | 2DGS [18] | 0.48 | 0.91 | 0.39 | 0.39 | 1.01 | 0.83 | 0.81 | 1.36 | 1.27 | 0.76 | 0.70 | 1.40 | 0.40 | 0.76 | 0.52 | 0.80 | 20m |
| | GOF [55] | 0.50 | 0.82 | 0.37 | 0.37 | 1.12 | 0.74 | 0.73 | 1.18 | 1.29 | 0.68 | 0.77 | 0.90 | 0.42 | 0.66 | 0.49 | 0.74 | 2h |
| | GS2Mesh [45] | 0.59 | 0.79 | 0.70 | 0.38 | 0.78 | 1.00 | 0.69 | 1.25 | 0.96 | 0.59 | 0.50 | 0.68 | 0.37 | 0.50 | 0.46 | 0.68 | 20m |
| | PGSR [6] | 0.36 | 0.57 | 0.38 | 0.33 | 0.78 | 0.58 | 0.50 | 1.08 | 0.63 | 0.59 | 0.46 | 0.54 | 0.30 | 0.38 | 0.34 | 0.52 | 30m |
| | GS-Pull [57] | 0.51 | 0.56 | 0.46 | 0.39 | 0.82 | 0.67 | 0.85 | 1.37 | 1.25 | 0.73 | 0.54 | 1.39 | 0.35 | 0.88 | 0.42 | 0.75 | 22m |
| | **Eve3D (Ours)** | 0.33 | 0.47 | 0.32 | 0.33 | 0.73 | 0.58 | 0.44 | 1.00 | 0.62 | 0.54 | 0.43 | 0.45 | 0.29 | 0.38 | 0.32 | 0.48 | 15m |

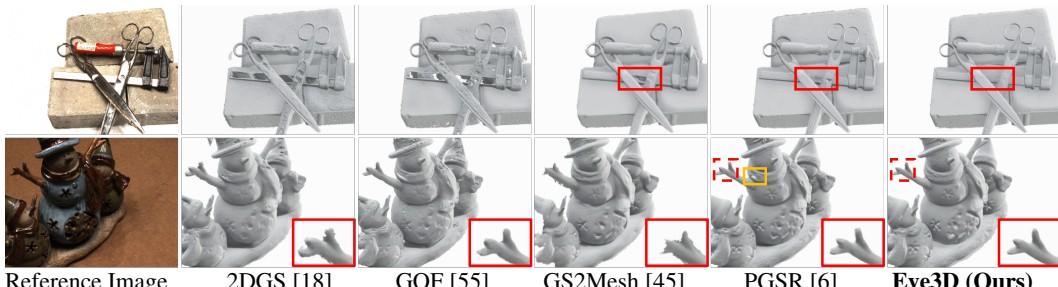

| Reference Image | 2DGS [18] | GOF [55] | GS2Mesh [45] | PGSR [6] | **Eve3D (Ours)** |

Figure 4: **Qualitative Comparison on the DTU Dataset.** Visual comparison of 3D meshes reconstructed by our approach versus previous methods.

**Implementation Details.** Training Eve3D occurs in two phases: first, we pre-train a vanilla 3DGS model using [13], which we use to render pseudo stereo views that feed into [44] to predict dense depth maps $D^*$ as priors. We use the pretrained model without fine-tuning, ensuring zero-shot generalization with no overlap between its training data and our evaluation datasets. We employ a left-right consistency check with a 3-pixel threshold to estimate confidence masks $M^c$. We set $T_{joint}$ to 7000, introduce depth prior supervision after the first 500 iterations, and apply local bundle adjustment from the beginning of training. We also built the Eve3D-*fast* variant, reducing the total iterations from 30k to 5k and setting $T_{joint}$ to 1000. Following PGSR [6], we set $\lambda_{dn} = 0.015$, $\lambda_s = 100.0$, and $\lambda_c = 1.0$. For our proposed losses, we use $\lambda_{prior} = 0.05$, $\lambda_{pull} = 0.05$, and $\lambda_{lba} = 0.15$. These hyperparameters remain fixed across all datasets without tuning. Finally, we use $K = 4$ neighboring views for local bundle adjustment across all experiments.

## 4.2 Evaluation Against State-of-the-Art

**Tanks and Temples.** The Tanks and Temples dataset [24] collects scenes in a surround manner, comprising six diverse environments that include both indoor and outdoor scenarios with varying scales and lighting conditions. Tab. 1 presents our quantitative evaluation based on F1-score (the higher the better). Eve3D achieves superior performance with an average of 0.58, outperforming both implicit methods and other explicit approaches. Our method achieves the highest ranking in five scenes (Barn, Caterpillar, Courthouse, Meeting Room, and Truck) and ranks second in one scene (Ignatius). The most significant improvements appear in challenging scenarios: Eve3D achieves an F1 Score of 0.46 in Meetingroom (compared to 0.33 from PGSR) and 0.35 in Courthouse (compared to 0.28 from the best competitors). Despite these substantial quality improvements, Eve3D (trained for 30K iterations) maintains a reasonable 1.2-hour total training time (including 3DGS pretraining, stereo rendering, and depth prediction) – significantly faster than implicit methods (>24h) while delivering superior reconstruction quality. Our Eve3D-*fast* variant, trained for only 5K iterations (20 minutes total time), still achieves second-best reconstruction quality with an average F1-score of 0.56, setting an unprecedented trade-off between accuracy and speed. Fig. 3 shows Eve3D addresses standard 3DGS limitations in scenes with varied lighting and complex architecture, producing more accurate flat surfaces and detailed structures in both indoor and outdoor settings.

Table 3: **Quantitative Comparisons of Novel View Synthesis on the Mip-NeRF360 Dataset.** ▨, ▨, ▨ indicate the absolute, second, and third bests, respectively.

| | Outdoor Scenes | | | Indoor Scenes | | |
|---|---|---|---|---|---|---|
| | PSNR ↑ | SSIM ↑ | LPIPS ↓ | PSNR ↑ | SSIM ↑ | LPIPS ↓ |
| NeRF [32] | 21.46 | 0.458 | 0.515 | 26.84 | 0.790 | 0.370 |
| Deep Blending [17] | 21.54 | 0.524 | 0.364 | 26.40 | 0.844 | 0.261 |
| Instant NGP | 22.90 | 0.566 | 0.371 | 29.15 | 0.880 | 0.216 |
| MERF | 23.19 | 0.616 | 0.343 | 27.80 | 0.855 | 0.271 |
| BakedSDF | 22.47 | 0.585 | 0.349 | 27.06 | 0.836 | 0.258 |
| MipNeRF360 | 24.47 | 0.691 | 0.283 | 31.72 | 0.917 | 0.180 |
| SuGaR [16] | 22.93 | 0.629 | 0.356 | 29.43 | 0.906 | 0.225 |
| 3DGS [22] | 24.64 | 0.731 | 0.234 | 30.41 | 0.920 | 0.189 |
| 2DGS [18] | 24.34 | 0.717 | 0.246 | 30.40 | 0.916 | 0.195 |
| GOF [55] | 24.76 | 0.742 | 0.225 | 30.80 | 0.928 | 0.167 |
| PGSR [6] | 24.76 | 0.752 | 0.203 | 30.36 | 0.934 | 0.147 |
| **Eve3D (Ours)** | 24.99 | 0.758 | 0.203 | 30.42 | 0.930 | 0.157 |

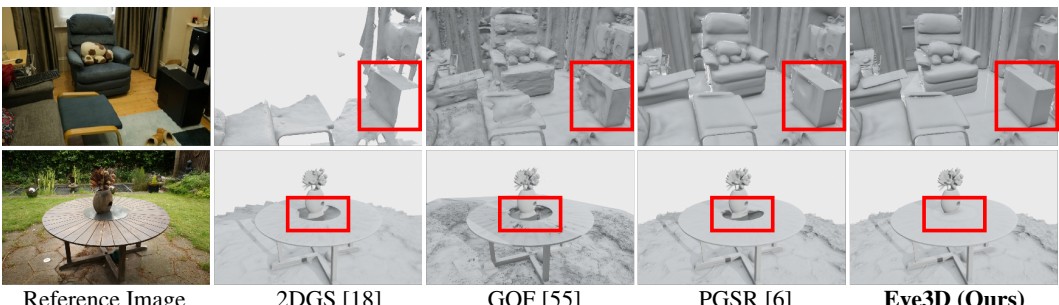

| Reference Image | 2DGS [18] | GOF [55] | PGSR [6] | **Eve3D (Ours)** |

Figure 5: **Qualitative Comparison on the Mip-NeRF360 Dataset.** Visual comparison of 3D meshes reconstructed by our approach versus previous methods.

**DTU.** Tab. 2 shows our Chamfer Distance evaluation (lower is better) on DTU [20]. Eve3D achieves the best average performance with a score of 0.48, outperforming both the best implicit method (Neuralangelo at 0.61) and the previous best explicit approach (PGSR at 0.52). Our method demonstrates remarkable consistency, achieving the best performance in any scenes except one, where it is the second-best. Importantly, our method maintains an efficient 15-minute total training time (including all preprocessing steps), comparable to vanilla 3DGS while delivering largely superior reconstruction quality. Qualitative comparisons in Fig. 4 show that while existing 3DGS methods produce good object meshes, they struggle with sparse viewpoints and inconsistent multi-view images. Our method enhances robustness in these difficult scenarios, yielding more complete, detailed meshes with better preserved fine structures.

**Mip-NeRF360.** For validating rendering quality, we evaluate on the Mip-NeRF360 dataset [2] following 3DGS's standard protocol. Tab. 3 shows our results using standard metrics. Eve3D achieves excellent synthesis across outdoor and indoor scenes. For outdoor scenes, Eve3D delivers best performance on all metrics: PSNR (24.99), SSIM (0.758), and LPIPS (0.203), outperforming both NeRF-based approaches and recent 3DGS methods. In indoor scenes, instead, our method achieves the second-highest SSIM (0.930) and LPIPS (0.157) scores, while maintaining competitive PSNR (30.42, third-best after MipNeRF360 and GOF). This demonstrates that our geometry-aware optimization also enhances rendering quality. Fig. 5 further highlights this qualitatively.

## 4.3 Ablation Study

We ablate the key components of Eve3D on the Tanks and Temples [24] dataset in Tab. 4.

**Single-view Prior Loss.** Due to the lack of explicit geometric constraints, the baseline model struggles to reconstruct accurate surfaces relying solely on RGB supervision. When a single-view depth prior is introduced to constrain the rendered depth and normals of 3D Gaussians, the surface reconstruction performance improves from 0.340 to 0.463. We also ablate the use of single-view prior loss in a joint optimization setting, where the prior depth maps are treated as learnable parameters. This leads to a further improvement from 0.523 to 0.539. Please refer to the supplementary material for experiments using different types of single-view priors derived from other vision models.

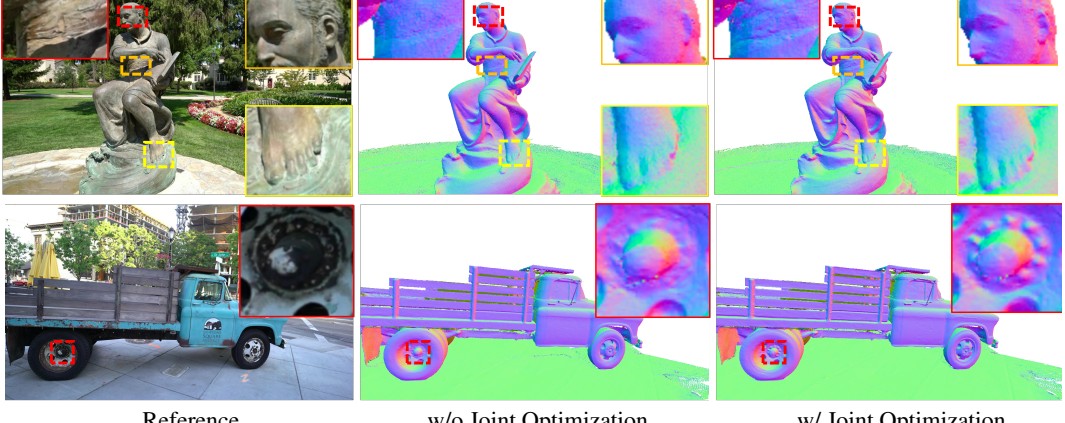

|  | Reference | w/o Joint Optimization | w/ Joint Optimization |

Figure 6: **Qualitative Mesh Comparison.** Results without (w/o) and with (w/) joint optimization.

**Local Bundle Adjustment.** Local bundle adjustment applies explicit multi-view consistency constraints on the rendered depth and prior depth maps. Solely with local bundle adjustment, the performance is improved significantly from 0.340 to 0.523. In this configuration, prior depths guide the multi-view consistency but remain non-learnable. Enabling joint optimization, which treats priors as learnable parameters $\hat{D}$, yields further improvement from 0.523 to 0.539, demonstrating that allowing the model to refine priors through backpropagation enhances both prior quality and 3DGS reconstruction. When using both single-view prior loss and local bundle adjustment, jointly optimizing 3DGS and depth priors also has significant effects, improving the F1 score from 0.553 to 0.581.

Table 4: **Component Contribution.** Evaluation of each module's effect on reconstruction quality.

| Single-view Prior Loss | Local Bundle Adjustment | Joint Optimization | P ↑ | R ↑ | F1 ↑ |
|:---:|:---:|:---:|:---:|:---:|:---:|
|  |  |  | 0.297 | 0.418 | 0.340 |
| ✓ |  |  | 0.431 | 0.519 | 0.463 |
| ✓ |  | ✓ | 0.437 | 0.523 | 0.467 |
|  | ✓ |  | 0.483 | 0.594 | 0.523 |
|  | ✓ | ✓ | 0.504 | 0.605 | 0.539 |
| ✓ | ✓ |  | 0.531 | 0.595 | 0.553 |
| ✓ | ✓ | ✓ | 0.553 | 0.631 | 0.581 |

**Joint Optimization.** We analyze the impact of each joint optimization component in Tab 5. The local bundle adjustment pre-training plays an important role in initializing multi-view consistent priors; removing it leads to a performance drop from 0.581 to 0.574. Disabling the confidence mask update causes a slight drop to 0.578, highlighting the benefit of including more prior regions—when validated by geometry checks—during joint optimization. Fig 6 compares results with and without joint optimization. Jointly optimizing Gaussians and priors enables reconstruction of fine details, while disabling it leads to over-smoothed surfaces due to excessive reliance on prior supervision. Furthermore, this joint strategy is also especially effective for recovering geometry details that are very ambiguous from single-view visual clues, such as the dark areas in the Truck.

Table 5: **Component Contribution – Joint Optimization.** Evaluation of each module's effect.

| Methods | P ↑ | R ↑ | F1 ↑ |
|:---|:---:|:---:|:---:|
| Joint Optimization (Full) | 0.553 | 0.631 | 0.581 |
| w/o LBA Pre-training | 0.549 | 0.618 | 0.574 |
| w/o Confidence Mask Update | 0.552 | 0.626 | 0.578 |

In Tab. 6, we compare the FoundationStereo [44] prior depths before and after joint optimization with Eve3D. We evaluate depth accuracy on the Tanks and Temples dataset using ground-truth depth maps provided by the RobustMVD benchmark. Among the four scenes available in RobustMVD, we use the three that overlap with our experimental setup: Barn, Courthouse, and Ignatius.

Since the camera poses estimated by COLMAP are not aligned with the ground-truth poses—leading to inconsistencies in depth scale—we perform mesh-to-mesh alignment between the reconstructed and ground-truth geometry to obtain accurate scaling and alignment information. We then compute the relative depth error to quantify performance.

As shown in the comparison, in all evaluated scenes, the Eve3D rendering results consistently outperform the initial priors in depth accuracy. Furthermore, our proposed joint optimization strategy improves the accuracy of the initial depth priors.

Table 6: **Impacts of Joint Optimization.** We evaluate the accuracy of the learned depth priors before and after joint optimization using ground-truth depth maps from the Tanks and Temples dataset, provided by the Robust Multi-view Depth (RobustMVD) benchmark [36]. Relative error rates are reported to quantify the improvement in depth estimation.

| Methods | Barn (%) | Courthouse (%) | Ignatius (%) |
|---|---|---|---|
| FoundationStereo [44] | 1.84 | 12.25 | 1.43 |
| Optimized FoundationStereo | 1.51 | 11.96 | 1.00 |
| Eve3D Depth | 1.48 | 11.79 | 0.80 |

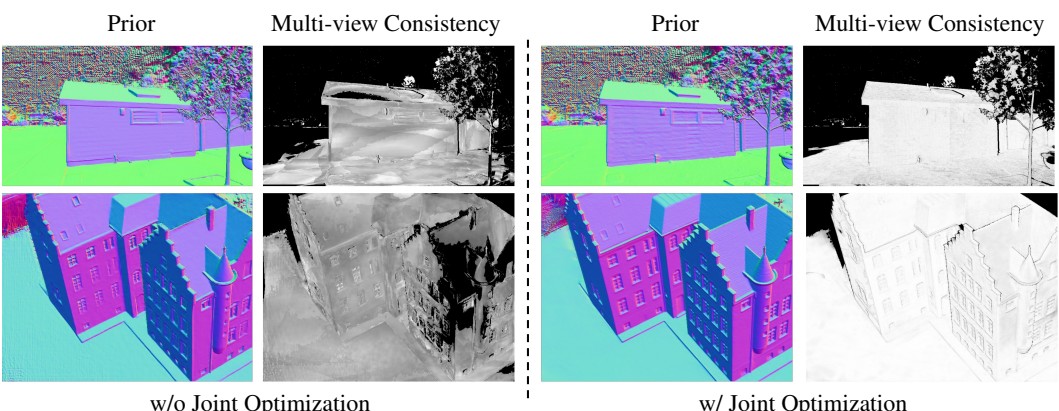

Figure 7: **Visualization of the multi-view consistency without and with the joint optimization.** The consistency of current-view depth priors is measured by computing the pixel-wise reprojection differences with the eight nearest neighbor views, followed by a consistency measurement using an exponential decay function, $\exp(-d)$, where d denotes the reprojection error. This weighting emphasizes geometrically consistent regions and suppresses unreliable estimates. Joint optimization significantly improves the multi-view consistency.

Additionally, in Fig 7, we highlight another key property of the priors: multi-view consistency. While initial priors provide reasonable but coarse estimates of depth and normals, they often exhibit inconsistency across different viewpoints. This inconsistency introduces noise, which can degrade the quality of supervision during Gaussian optimization. With our proposed joint optimization strategy, the multi-view consistency of priors is significantly improved, resulting in more stable and accurate supervision signals.

## 5  Conclusion

We presented Eve3D, a novel framework for dense surface reconstruction based on 3DGS. Our approach jointly optimizes both self-derived stereo depth priors and the 3DGS representation, establishing a mutually beneficial relationship in which each component improves the other. Our local bundle adjustment strategy ensures global consistency across view-overlapping frames, effectively compensating for the local supervision limitations inherent in 3DGS. Extensive experiments on Tanks & Temples, DTU, and Mip-NeRF360 demonstrate that Eve3D achieves state-of-the-art performance in both surface reconstruction and novel view synthesis, while training in as little as 15-20 minutes for our fast version, and ~1 GPU hours for our base approach.

**Limitations.** Eve3D sets a new state-of-the-art, yet with some trade-offs. Its primary constraint is the reliance of a vision foundation model for stereo depth estimation. This choice is motivated by the unpaired accuracy of the estimated depth priors compared to alternative monocular [48] or multi-view stereo [19] solutions, as discussed in the supplementary material. However, it requires rendering stereo images from the model itself – an overhead that could be avoided if a multi-view stereo foundation model could achieve comparable accuracy.

## Acknowledgments and Disclosure of Funding

This work is supported by Rawmantic AI, and the National Natural Science Foundation of China 62276016, 62372029.

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

# Appendix for "Eve3D: Elevating Vision Models for Enhanced 3D Surface Reconstruction via Gaussian Splatting"

## Overview

This appendix contains supplementary material that supports and extends the findings presented in the main paper. We begin in Sec. A with detail descriptions of our experimental setup. Sec. B provides further analysis, including ablation studies to better understand different components of our approach. Additional qualitative results of 3D reconstructions produced by Eve3D are presented in Sec. C. Finally, in Sec. D, we reflect on the broader impact of our methodology.

## A    Details of Experimental Setting

### A.1    Datasets

**DTU.** The DTU dataset provides ground-truth point clouds for evaluating object-level reconstruction quality. Following prior works [18, 55, 6], we use 15 scans (24, 37, 40, 55, 63, 65, 69, 83, 94, 102, 106, 110, 114, 118, and 122) to assess surface reconstruction performance. In our experiments, all images from each scan are used, downsampled to half resolution for training.

**Tanks and Temples.** The Tanks and Temples dataset includes ground-truth points for evaluating surface reconstruction in both indoor and outdoor scenes. In line with previous studies [18, 55, 6], we conduct experiments on six scenes: Barn, Caterpillar, Courthouse, Ignatius, Meetingroom, Truck. For each scene, we use all available images, downsampled to half resolution for training.

**Mip-NeRF360.** Since Mip-NeRF360 does not provide ground-truth points for surface reconstruction evaluation, we instead use it to evaluate novel view synthesis performance. We adopt the standard train/test splits from prior works [18, 55, 6]. For outdoor scenes (bicycle, flowers, garden, stump, treehill), images are downsampled to quarter resolution. For indoor scenes (bonsai, counter, kitchen, room), images are downsampled to half resolution, consistent with previous studies [18, 55, 6]. For mesh reconstruction visualizations, we train models using only the training split images.

### A.2    Implementations

**Hyperparameters.** Our base model adopts the plane depth definitions [6] to render depth. We constrain the shortest axis scale of Gaussians to zero to make Gaussians as close to planes. We adopt a depth-normal consistency loss [18, 55, 6] to encourage the consistent representations of rendered depth and normal vectors. The learnable prior depth maps are initialized using predictions from a depth estimation model and optimized with a learning rate of $(1 \times 10^{-4})$. For depth map initialization, we sample 500,000 points for DTU scans and 1,000,000 points for both Tanks and Temples scenes and the Mip-NeRF360 dataset.

**Eve3D.** We train Eve3D for a total of 30,000 iterations. Prior depth supervision is introduced starting from iteration 500. We set the $T_{joint}$ to 7000. The shortest axis scale loss is applied from the beginning of training. The depth-normal consistency is activated starting at iteration 7000. The densification process for 3D Gaussians begins at iteration 500 and ends at iteration 15,000.

**Eve3D-*fast*.** We train Eve3D-*fast* for a total of 5,000 iterations. Prior depth supervision is introduced starting from iteration 500. We set $T_{joint} = 1000$. The shortest axis scale loss is applied from the beginning of training. The depth-normal consistency is enabled from iteration 1000. The densification of 3D Gaussians begins at the 500 iteration and concludes at iteration 4000.

**Mesh Extraction.** We render depth maps from the 3D Gaussians and apply Truncated Signed Distance Function (TSDF) fusion to extract surface meshes. For scenes captured with front-facing cameras (DTU), we use unbounded mesh extraction and set the voxel size to 0.002. For scenes captured by surround-view cameras (e.g., Tanks and Temples, Mip-NeRF360), we use bounded mesh extraction, where the voxel size is set to the maximum scene extent divided by 2048. For indoor scenes, scene bounds are estimated from camera trajectories, while for outdoor scenes, they are estimated from the reconstructed point clouds.

Table 7: **Direct Comparisons between Eve3D and GS2Mesh.** Methods are trained with mini-splatting2 to render stereo views and FoundationStereo to predict depth maps.

| Methods | Barn | Caterpillar | Courthouse | Ignatius | Meetingroom | Truck | Mean ↑ | Time |
|---------|------|-------------|------------|----------|-------------|-------|--------|------|
| GS2Mesh [45] | 0.51 | 0.27 | 0.08 | 0.61 | 0.19 | 0.41 | 0.35 | 12 m |
| Eve3D-*fast* (Ours) | 0.69 | 0.44 | 0.34 | 0.82 | 0.41 | 0.62 | 0.56 | 20 m |
| Eve3D (Ours) | 0.70 | 0.48 | 0.35 | 0.83 | 0.46 | 0.66 | 0.58 | 1.2 h |

Table 8: **Comparisons to PGSR with Depth Priors.** We train PGSR with the FoundationStereo initialization and supervision, which is the same to the supervision used in Eve3D. The difference is that Eve3D uses the Prior-involved bundle adjustment with joint optimization, while PGSR uses multi-view consistency between neighbor-view rendering results to maintain the multi-view consistency. With the same depth prior, Eve3D shows significantly better convergence than PGSR.

| Total | PGSR + FoundationStereo | | | Eve3D (Ours) | | |
|-------|-------------|-----------|------------|-------------|-----------|------------|
| Iterations | Precision ↑ | Recall ↑ | F1 Score ↑ | Precision ↑ | Recall ↑ | F1 Score ↑ |
| 3k | 0.494 | 0.531 | 0.485 | 0.500 | 0.570 | 0.525 |
| 5k | 0.492 | 0.574 | 0.519 | 0.532 | 0.600 | 0.555 |
| 10k | 0.505 | 0.586 | 0.532 | 0.546 | 0.611 | 0.568 |
| 30k | 0.552 | 0.609 | 0.571 | 0.553 | 0.631 | 0.581 |

**Overlapping Score.** Following [51], for a reference view $V_i$, we compute the overlapping score $s(i,j) = \sum_X \eta(\theta_{ij}(X))$ for its neighboring view $V_j$, and $X$ is a 3D point which is observed by both views $V_i$ and $V_j$. In detail, $\theta_{ij}(X) = (180/\pi) \arccos((t_i - X) \cdot (t_j - X))$ is the baseline angle and $t$ represents the camera center. $\eta(\cdot)$ is piece-wise Gaussian function that favors a certain baseline angle $\theta_0$:

$$\eta(\theta) = \begin{cases} \exp\left(-\frac{(\theta-\theta_0)^2}{2\sigma_1^2}\right), & \text{if } \theta \leq \theta_0 \\ \exp\left(-\frac{(\theta-\theta_0)^2}{2\sigma_2^2}\right), & \text{if } \theta > \theta_0 \end{cases} . \tag{16}$$

where $\theta_0$, $\sigma_0$ and $\sigma_1$ are hyper-parameters and are set to 5, 1, and 10 respectively.

## B  Additional Analysis

### B.1  Direct Comparisons with Improved GS2Mesh

Eve3D leverages the rendering capabilities of 3DGS to generate stereo pairs and infer depth priors—similar in spirit to GS2Mesh [45]. However, while GS2Mesh directly uses stereo depth maps to reconstruct the meshes, our apporach treats stereo depth maps as priors, which are then jointly optimized along with the 3D Gaussian via our proposed framework.

Although the original GS2Mesh significantly underperforms compared to Eve3D in terms of reconstruction accuracy (see Tab. 2 in the main paper), the reader might argue that the discrepancy could be due to the use of a different stereo backbone-DLNR [59]. Therefore, to fully assess the superiority of our methodology over the direct fusion of stereo priors, we re-implement GS2Mesh using the same settings as Eve3D.

Accordingly, we provide a new comparison between GS2Mesh and Eve3D in Tab. 7 on the Tanks and Temples dataset. In this experiment, both methods use Mini-Splatting [13] to render pseudo stereo views and FoundationStereo [44] to estimate depth maps. Even with such high-quality priors, GS2Mesh struggles to reconstruct accurate surfaces—particularly in complex scenes like Meetingroom and Courthouse. In contrast, our Eve3D-fast, with only eight minutes of additional optimization, achieves significantly better reconstruction quality.

Table 9: **Ablation Study.** Impact of vision model choice.

| Prior Source | Eve3D (Ours) | Precision ↑ | Recall ↑ | F1 Score ↑ |
|---|---|---|---|---|
| FoundationStereo [44] | ✗ | 0.431 | 0.519 | 0.463 |
| FoundationStereo [44] | ✓ | 0.553 | 0.631 | 0.581 |
| Stereo Anywhere [3] | ✗ | 0.410 | 0.470 | 0.431 |
| Stereo Anywhere [3] | ✓ | 0.533 | 0.600 | 0.555 |
| MVSAnywhere [19] | ✗ | 0.430 | 0.517 | 0.462 |
| MVSAnywhere [19] | ✓ | 0.506 | 0.578 | 0.532 |
| MVSFormer [4] | ✗ | 0.450 | 0.548 | 0.483 |
| MVSFormer [4] | ✓ | 0.493 | 0.598 | 0.528 |
| OMNI-DC [64] | ✗ | 0.298 | 0.390 | 0.330 |
| OMNI-DC [64] | ✓ | 0.448 | 0.540 | 0.479 |

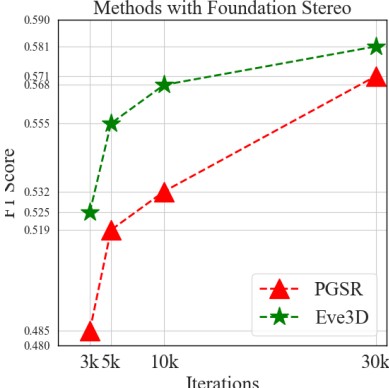

Figure 8: **Comparisons of PGSR and Eve3D with FoundationStereo priors – convergence speed.**
.

## B.2 Comparisons to PGSR with Depth Prior

PGSR [6] sets a strong baseline for surface reconstruction with the proposed multi-view consistency based on rendered results. Compared to PGSR, Eve3D uses the prior-depth involved bundle adjustment to enhance the multi-view consistency, incorporating more than one neighbor view in one training loop. Moreover, Eve3D leverages bundle adjustment not only to refine the 3D Gaussians but also to optimize the depth priors themselves. When the initial priors are reasonably accurate at a coarse level, they can be quickly refined into multi-view consistent priors through local bundle adjustment. This leads to better convergence behavior compared to enforcing multi-view constraints directly on rendering outputs, as done in PGSR. As shown in Table8 and Figure8, when using the same FoundationStereo priors, Eve3D achieves significantly faster and more stable convergence than PGSR.

Table 10: **Ablation Study.** Impacts of baseline length.

| Baseline Length | 3 % camera extent | 7 % camera extent | 10 % camera extent |
|---|---|---|---|
| F1 score ↑ | 0.580 | 0.581 | 0.581 |

## B.3 Additional Ablation Study on Method Components

**Prior Depth Types.** We evaluate the generalizability of our method across alternative sources for depth priors, replacing those obtained from FoundationStereo applied to rendered stereo images with different approaches: i) using the Stereo Anywhere model [3]; ii) using depth maps from multi-view stereo (MVSAnywhere [19] and MVSFormer [4]) or iii) a depth completion network (OMNI-DC [64]) applied to sparse depth points extracted from COLMAP. All these methods predict depth maps

Table 11: **Ablation study.** Impact of the neighbors in local bundle adjustment.

| Number of Neighbors | Precision ↑ | Recall ↑ | F1 Score ↑ | Training Time |
|---|---|---|---|---|
| 1 | 0.544 | 0.625 | 0.573 | 50 m |
| 2 | 0.551 | 0.627 | 0.578 | 1 h |
| 4 | 0.553 | 0.631 | 0.581 | 1.2 h |
| 8 | 0.554 | 0.631 | 0.582 | 1.5 h |

at the correct metric scale, although through different working principles: stereo models estimate disparity maps from stereo images rendered using intrinsics and extrinsics at the same scale as the pretrained 3DGS, then triangulate depth using the known focal length and baseline; multi-view stereo methods exploit the same camera poses used to optimize 3DGS, thus predicting depth at consistent scale; depth completion models densify sparse COLMAP points used to initialize 3DGS, maintaining their metric scale. Despite variations in depth accuracy across these sources, our joint optimization consistently improves reconstruction performance (Table 9), demonstrating robustness to different depth initializations and strong generalization. We emphasize that all foundation models used in our experiments are applied zero-shot without fine-tuning.

We also highlight how, at the current stage, rendering stereo images to extract priors through FoundationStereo [44] represents the optimal choice; nonetheless, we don't exclude that future advances in multi-view stereo or depth completion may lead to stronger models, thus making Eve3D no longer require rendering stereo images to get priors. Confirming this hypothesis in future research would allow for further improve Eve3D performance – as it can be seamlessly integrated even with future, more advanced networks.

**Virtual Camera Baseline Length.** For stereo pair generation, we set the baseline length to 7% of the scene radius across all experiments. To assess the sensitivity of our method to this choice, we evaluate different baseline lengths in Table 10. The results demonstrate that our method is robust to baseline selection, with F1 scores remaining consistent (0.580-0.581) across baseline lengths ranging from 3% to 10% of the camera extent. This robustness stems from our joint optimization and local bundle adjustment, which enforce multi-view consistency constraints that naturally compensate for variations in initial stereo depth estimates.

**Number of Neighbors in Local Bundle Adjustment.** We study the impact of the number of neighboring views used in local bundle adjustment in Table 11. As the number of neighbors increases, the training time also grows due to the additional computation. At the same time, the inclusion of more diverse viewing angles in each optimization step enhances the geometric accuracy of both the 3DGS representation and the optimized depth priors. However, beyond a certain point, the benefit of adding more neighbors saturates. This is because additional views with weaker co-visibility relationships contribute limited new information, resulting in diminishing returns in geometric improvement.

### B.4 Training Time Breakdown

All training times reported in the main paper include the complete pipeline: 3DGS pretraining, stereo pair rendering, FoundationStereo predictions, and final 3DGS training. We provide a detailed breakdown of preprocessing and training times for transparency.

**Tanks and Temples:** On average, it takes 4 minutes to pretrain 3DGS with Mini-Splatting [13], 8 minutes for stereo pair rendering and FoundationStereo depth predictions, and 60 minutes for final Eve3D training (8 minutes for Eve3D-*fast*). The total time is therefore 1.2 hours for Eve3D and 20 minutes for Eve3D-*fast*.

**DTU:** On average, it takes 3 minutes to pretrain 3DGS with Mini-Splatting, 4 minutes for stereo pair rendering and FoundationStereo predictions, and 8 minutes for final training. The total time is 15 minutes.

## C  Additional Visualization Results

We provide additional qualitative results of Eve3D in Fig. 9, 10, and 11, which illustrate the surface reconstructions on the Tanks and Temples, DTU, and Mip-NeRF360 datasets, respectively.

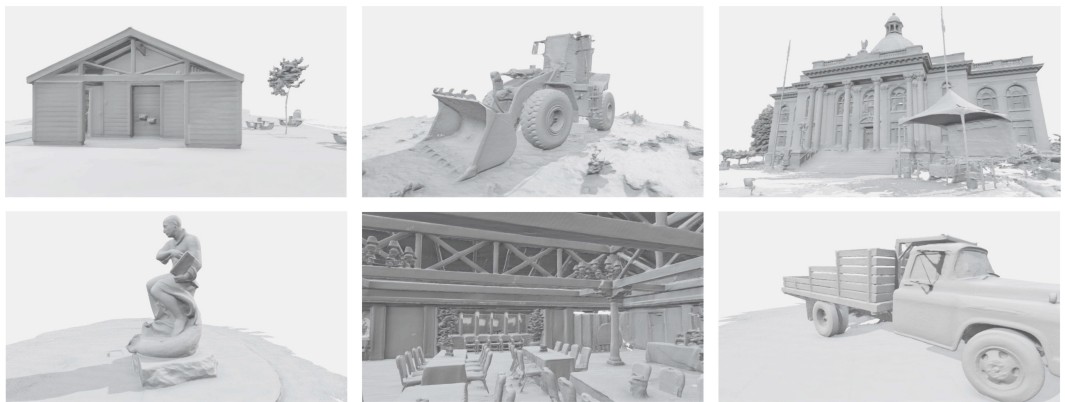

Figure 9: **Qualitative Visualizations on the Tanks and Temples Dataset.**

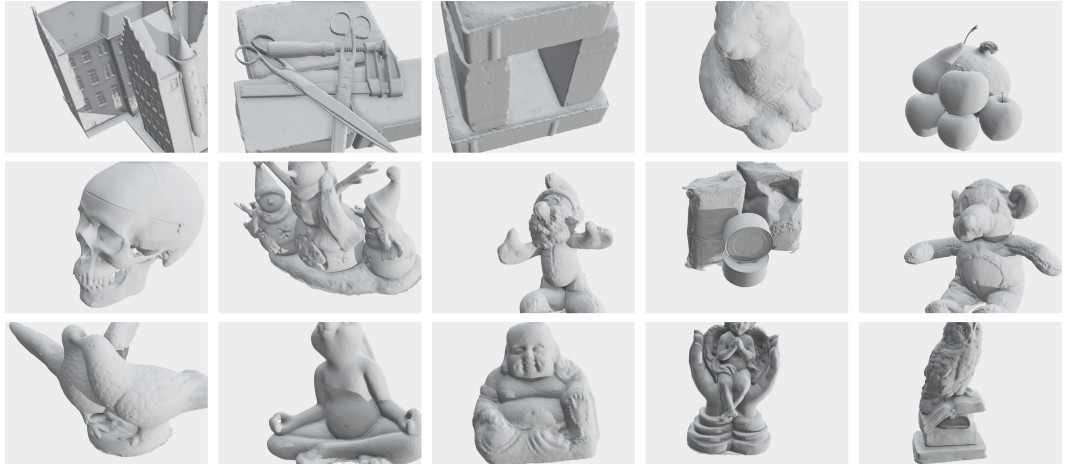

Figure 10: **Qualitative Visualizations on the DTU Dataset.**

# D    Broader Impact Statement

Eve3D sets a new state-of-the-art in 3D surface reconstruction, achieving unprecedented accuracy with a very low time and hardware budget.

On the one hand, Eve3D has the potential to accelerate progress across several high-level applicative domains, including augmented/virtual reality, robotics, autonomous navigation/interaction with the environment, and 3D content creation. The accuracy-speed trade-off achieved by Eve3D could represent a strong opportunity to democratize access to high-quality 3D modeling, by significantly lowering the entry barriers for researchers, educators, or any independent developers. Furthermore, a faster convergence speed also translates into a reduced carbon footprint associated to 3D reconstruction.

On the other hand, the possibility of producing higher-quality 3D models also comes with ethical considerations. These latter could be misused for applications such as surveillance or other privacy-infringement purposes. However, we argue Eve3D is not designed to handle dynamic objects/subjects during the reconstruction process, thus making it unsuited for processing casually collected videos where subjects may appear without their explicit consent.

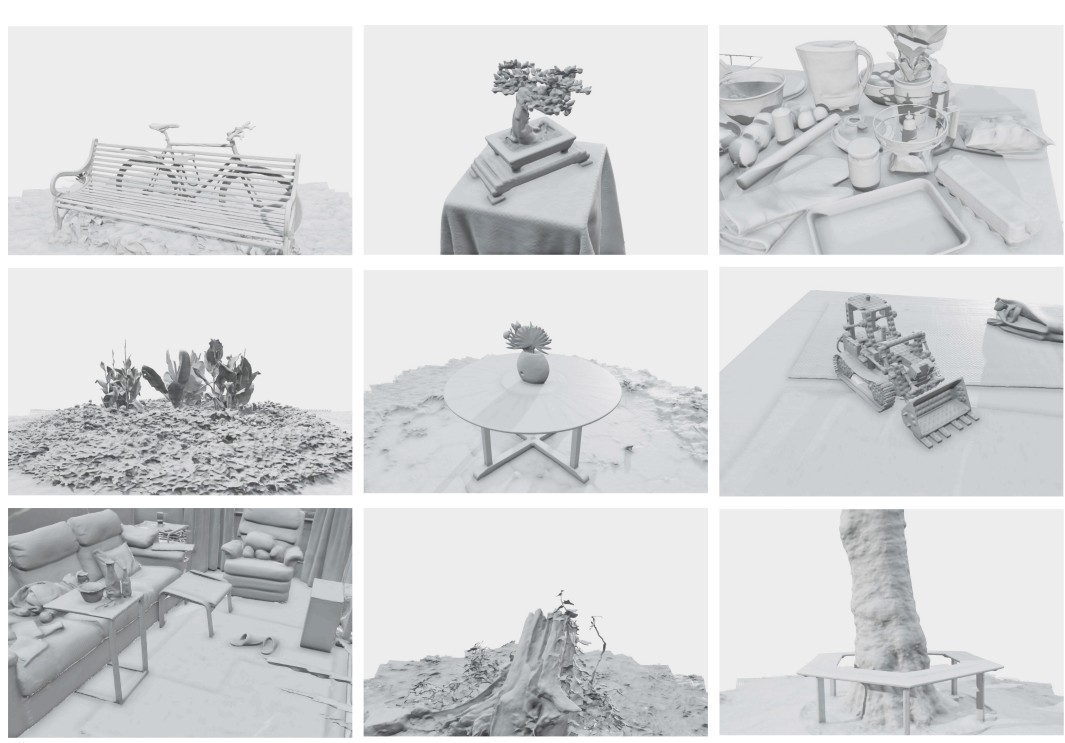

Figure 11: **Qualitative Visualizations on the Mip-NeRF360 Dataset.**

