# OpenReview forum: "Eve3D: Elevating Vision Models for Enhanced 3D Surface Reconstruction via Gaussian Splatting"
_NeurIPS.cc/2025/Conference — NeurIPS 2025 poster_

### Official Review · Reviewer_wAXX · 2025-06-29

**Clarity:** 3
**Significance:** 3
**Originality:** 2
**Rating:** 4
**Confidence:** 5

**Summary:**

Method:
1) run stereo matching on pairs of images with existing network [37]
2) include depth priors into 3dgs reconstruction and run joint optimization / bundle adjustment to obtain 3dgs scene representation

Core contribution is the use of a specific stereo matching network for better 3dgs reconstructions.

**Questions:**

- Is the stereo matching network trained on the respective datasets? How do you make sure there is no overlap in the train/test split?
- How much do the priors help vs how much the joint optimization? I.e., I would like to see another entry in Table 1 and 2.
- What if the baselines such as 2DGS would have the prior as a helper? I understand that there is some engineering to be done there as we well which this method contributes but at the same time the loss formulation is not that complicated either.
- How would the surface look like if just the respective stereo depth maps were fused into a joint 3D scene representation (i.e., no 3dgs at all)? E.g., using something like a SDF-based reconstruction framework such as KinectFusion variants or Poisson Surface Reconstruction.

I'm generally on the fence so I would like to see how the authors respond to the questions above.

**Ethical Concerns:**

["NO or VERY MINOR ethics concerns only"]

**Final Justification:**

I thank the authors for their rebuttal and additional comments. In combination of reading the other reviews, I would like to keep my original rating and I'm leaning positive towards the paper.

**Limitations:**

yes, there is a limitation section.

**Paper Formatting Concerns:**

looks good

**Quality:**

3

**Strengths And Weaknesses:**

Strengths:
- compelling results and comparisons against the baselines in Table 1 and 2.
- combining existing priors for surface reconstruction and 3DGS is always interesting.

Weakness:
- in the video, only a single scene is shown. The rest are mostly single objects which is a bit boring; the video rendering quality could be also better.
- algorithmically, the method is straightforward and it seems the majority from the improvements come from better engineering; e.g., which depth estimation prior to use etc.
- would be nice to see more ablations there how much that matters (also see below in the question section)

---

> ### Author Rebuttal · Authors · 2025-07-30
>
> ### **Responses to Reviewer wAXX** ###
>
> We thank the reviewers for their insightful feedback. Below we address the questions and concerns raised.
>
> ---
>
> >**Q1: In the video, only a single scene is shown. The rest are mostly single objects which is a bit boring; the video rendering quality could be also better.**
>
>
> We thank the reviewer for the constructive feedback. While the rebuttal roles do not allow us to include additional videos or supplementary materials, we fully agree with this suggestion and will enhance the video with more diverse scenes and improved rendering quality.
>
>
> ---
>
>
>
> >**Q2: Algorithmically, the method is straightforward and it seems the majority from the improvements come from better engineering; e.g., which depth estimation prior to use etc.**
>
>
> Thanks for the feedback. We'd like to declare that the significant improvement from our method comes from our novel and effective designs of parameterized learnable depth and local bundle adjustment for 3DGS-prior joint optimization, rather than the engineering on depth priors. As shown in Table 4 and Figure 6, our method can consistently improve the performance significantly, demonstrating the general criticalness and effectiveness of our strategies. Furthermore, Figure 8 in the appendix demonstrates that, under the same depth prior, our approach achieves faster convergence and higher reconstruction quality-highlighting the importance of how the priors used, not just which prior is selected.
>
>
> ---
>
> >**Would be nice to see more ablations there how much that matters (also see below in the question section)**
>
> Thanks for the feedback. Responses to the reviewer’s specific questions regarding these ablations are provided in detail below.
>
> ---
>
> >**Q3: Is the stereo matching network trained on the respective datasets? How do you make sure there is no overlap in the train/test split?**
>
> Thank you for pointing this out. The selected FoundationStereo network was not trained on our evaluation datasets. In fact, the DTU, Tanks \& Temples and Mip-NeRF360 datasets do not contain binocular stereo pairs, which makes them unsuitable for training a stereo matching network. FoundationStereo was trained on: Scene Flow, Sintel, CREStereo, FallingThings, InStereo2K and Virtual KITTI 2, none of which overlap with our test datasets. This ensures there is no data contamination and validates the zero-shot generalization capabilities of the stereo foundation model. The same principle applies to the other foundation models used in Table 9 (appendix), ensuring consistency and a fair evaluation protocol.
>
>
> ---
>
>
> >**Q4: How much do the priors help vs how much the joint optimization? I.e., I would like to see another entry in Table 1 and 2.**
>
>
> We agree that disentangling the contribution of the priors from that of the joint optimization is important, and we have already included such an analysis in different parts of the paper. In particular, Table 4 (main paper) compares row 2 (using the learned priors only, without joint optimization) to row 3 (with joint optimization), thereby isolating the contribution of the joint optimization step, showing an F-score improvement from 0.553 to 0.581. And the qualitative comparisons of the reconstructed mesh are shown in Figure 6. Additionally, Table 6 (appendix) quantifies how joint optimization improves the depth priors themselves.
>
> ---
>
>
> >**Q5: What if the baselines such as 2DGS would have the prior as a helper? I understand that there is some engineering to be done there as we well which this method contributes but at the same time the loss formulation is not that complicated either.**
>
>
> We thank the reviewer for raising this point. We run experiments with 2DGS by integrating the same depth priors (FoundationStereo) we use for Eve3D. Furthermore, we also implement our whole Eve3D pipeline assuming 2DGS as the backbone. We present the results achieved by these two frameworks on Tanks and Templetes in the following table, using the same hyperparameter settings as those in the main paper. Using FoundationStereo priors allows for improving mean F-score achieved by 2DGS from 0.30 to 0.45. Furthermore, by applying the whole Eve3D pipeline to 2DGS allows for further pushing the accuracy from 0.45 to 0.55, getting close to the one by our original Eve3D implementation.
> A similar experiment involving PGSR (the most accurate among the competitors) is shown in Table 8 and Figure 8 of the appendix. Even under identical depth supervision, PGSR still performs worse than our approach. This demonstrates that our improvements are not solely due to the use of priors, but also to how they are integrated and jointly optimized within our framework.
>
> **Table: Eve3D with 2DGS on Tanks and Temples datasets**
>
> | Method                  | Barn | Caterpillar | Courthouse | Ignatius | Meetingroom | Truck | **Mean ↑** |
> |-------------------------|------|-------------|-------------|----------|--------------|--------|-------------|
> | 2DGS                    | 0.36 | 0.23        | 0.13        | 0.44     | 0.16         | 0.26   | 0.30        |
> | 2DGS + FoundationStereo | 0.63 | 0.33        | 0.29        | 0.62     | 0.30         | 0.53   | 0.45        |
> | 2DGS + Eve3D            | **0.69** | **0.43**    | **0.31**    | **0.82** | **0.43**     | **0.61** | **0.55**    |
>
> ---
>
> >**Q6: How would the surface look like if just the respective stereo depth maps were fused into a joint 3D scene representation (i.e., no 3dgs at all)? E.g., using something like a SDF-based reconstruction framework such as KinectFusion variants or Poisson Surface Reconstruction.**
>
>
> Performing direct fusion of stereo depth maps into a joint 3D scene representation, as suggested, requires access to stereo pairs. However, the DTU, Tanks \& Temples, and Mip-NeRF360 datasets do not provide stereo pairs, which makes 3DGS rendering a necessary prerequisite step to generate these views. The GS2Mesh method, which we re-implement using our pipeline (Mini-Splatting + FoundationStereo), follows exactly this paradigm: after generating stereo pairs from a 3DGS model, it fuses the resulting depth maps via standard reconstruction (e.g., KinectFusion). We address this scenario in Table 7 (appendix) with a fair comparison: even when both GS2Mesh and our approach rely on the same high-quality stereo priors, GS2Mesh struggles in complex scenes, whereas our method achieves significantly better reconstructions. We attribute this improvement to our strategy of treating depth maps as learnable priors, jointly optimized with 3DGS, rather than relying on a fixed post-hoc fusion.
>
> ---

---

> > ### Comment · Reviewer_wAXX · 2025-08-04
> >
> > Thanks for providing the rebuttal. I'm still unclear on the last point (Q6). Obviously, it depends whether there are stereo pairs available in the respective test scenario, but there are plenty of MVS depth estimators that could be taken into account.
> >
> > My main question there is whether the 2DGS optimization is beneficial over traditional reconstruction pipelines that would directly operate on the the estimated depths. In particular, it would be interested to use exactly the *same* depth estimates that are fed into the current reconstruction approach; i.e., do the improvements come from the prior itself -potentially show ablation - ? Or is it the combination of 2DGS-like reconstruction formulations?

---

> > > ### Author Response · Authors · 2025-08-05
> > >
> > > Thanks for the comment. To address the concern that whether the improvements stem from the prior itself or from the combination with the 3DGS optimization, we apply the depth priors from MvsAnywhere to the TSDF-fusion reconstruction pipeline. We compare this pipeline to our Eve3D where the same MvsAnywhere depth priors are jointly optimized with 3DGS in the following table.
> > >
> > > As shown in the following table, optimizing depth priors with 3DGS yields signincant improvements than directly using depth priors for reconstruciton, demonstraiting the importances of the joint optimization formulations.  We think this is because the mvs prior models, during inference, each view depends on a set of independent neighbor views to predict depths, and therefore lack global geometric consistency. Additionally, the domain shifts between training and unseen scenes can degrade the generalization ability of such models. In contrast, our methods can effectively compensate for these limitations by integrating multi-view consistency during per-scene optimization. We will include the corresponding analysis and ablation results in the revised appendix.
> > >
> > >
> > > **Table: MvsAnywhere and Eve3d performances on Tanks and Templetes datasets.**
> > >
> > > | Method                  | Barn | Caterpillar | Courthouse | Ignatius | Meetingroom | Truck | Mean ↑ |
> > > |-------------------------|------|-------------|------------|----------|--------------|--------|--------|
> > > | MvsAnywhere             | 0.27 | 0.09        | 0.02       | 0.18     | 0.12         | 0.28   | 0.16   |
> > > | MvsAnywhere + Eve3D     | 0.68 | 0.40        | 0.28       | 0.80     | 0.40         | 0.62   | 0.53   |

---

> > > > ### Comment · Reviewer_wAXX · 2025-08-05
> > > >
> > > > Thanks for the quick reply. At this point, I have no further questions and would like to maintain my original rating.

---

### Official Review · Reviewer_Bwjv · 2025-07-01

**Clarity:** 4
**Significance:** 3
**Originality:** 3
**Rating:** 4
**Confidence:** 5

**Summary:**

This paper presents a 3DGS framework for surface reconstruction with joint optimization of depth priors. Specifically, a virtual camera is utilized to perform a standard binocular stereo matching in a multi-view setting. Besides, a prior-involved local bundle adjustment is proposed to further eliminate the noise in depth prediction.

**Questions:**

- How would the virtual camera affect the final results? Are the final results sensitive to the different choice of virtual cameras? An ablation study of different virtual cameras should be conducted.

- The generalizability of this method should be well studied, as depth estimation may fail to predict the right depth scale. Are the results sensitive to the initialization of depth estimation?

- Why did you choose depth priors other than normal or normal & depth?

**Ethical Concerns:**

["NO or VERY MINOR ethics concerns only"]

**Final Justification:**

Most of my concerns are addressed. The impact of different prior depths with Eve3D could be included in the supplementary materials.

**Limitations:**

Yes.

**Paper Formatting Concerns:**

None.

**Quality:**

3

**Strengths And Weaknesses:**

Strengths
- The paper introduces a novel approach that jointly optimizes pre-trained vision model priors and the 3D Gaussian Splatting (3DGS) backbone. This creates a mutually reinforcing cycle where priors enhance 3DGS quality, which in turn refines the priors, leading to significantly improved 3D reconstruction accuracy. The framework addresses the limitations of existing methods that rely on static priors.

- By introducing a local bundle adjustment strategy, the method overcomes the highly local supervision limitations of standard 3DGS pipelines. This approach enforces global consistency across co-visible frames during optimization, ensuring more accurate surface reconstruction and addressing challenges like textureless regions and repetitive patterns. The quantitative results on benchmarks like Tanks & Temples and DTU validate its effectiveness.

- Eve3D achieves superior results in surface reconstruction and novel view synthesis while maintaining fast convergence.

Weaknesses

- The framework relies on a vision foundation model for stereo depth estimation, which introduces an overhead of rendering stereo images from the model. While this improves accuracy, it limits the method’s independence and could be problematic in scenarios where such models are not available or efficient. The paper acknowledges this as a limitation but does not provide a robust alternative.

- Although the paper mentions that model-generated priors contain noise, the analysis of how different levels of prior noise affect reconstruction quality is insufficient. A more detailed ablation study on the impact of prior accuracy across various scenes would strengthen the understanding of the method’s robustness in noisy conditions.

- The experiments primarily focus on standard benchmarks, but the method’s performance in extreme scenarios (e.g., highly dynamic environments, low-light conditions, or extremely large-scale scenes) remains unaddressed. The paper does not provide evidence of its effectiveness beyond the tested datasets, limiting confidence in its broader applicability.

---

> ### Author Rebuttal · Authors · 2025-07-30
>
> ### **Responses to Reviewer Bwjv** ###
>
> We thank the reviewers for their insightful feedback. Below we address the questions and concerns raised.
>
> ---
>
> >**Q1: The framework relies on a vision foundation model for stereo depth estimation, which introduces an overhead of rendering stereo images from the model. While this improves accuracy, it limits the method’s independence and could be problematic in scenarios where such models are not available or efficient. The paper acknowledges this as a limitation but does not provide a robust alternative.**
>
>
> Our method aims to explore a general and effective way to integrate rapidly advancing geometric models into the 3D reconstruction pipeline—a direction we believe is of high value for the field. While this involves using external depth estimation models, it is a common and effective practice in many state-of-the-art computer vision tasks and enables better accuracy and convergence. Crucially, our framework is not tied to any particular depth model. As demonstrated in Appendix Table 9, it is compatible with various types of priors, and consistently improves reconstruction quality. This robustness stems from that the depth prior is parameterized and optimized during training, as well as a multi-view geometric check during optimization, allowing our system to correct or adapt to imperfect inputs. This enables the method to maintain a high-quality lower bound, even in challenging scenarios where the prior may be noisy or partially failing.
>
> In terms of efficiency, we emphasize that incorporating the stereo-based depth prior does not introduce additional overhead beyond our reported runtime. All timings, including stereo rendering, depth supervision, and 3DGS optimization, are already included in our measurements. The stereo pairs rendering is only needed once at the start of training. Moreover, the use of a depth prior not only improves final reconstruction quality but also accelerates convergence, providing a favorable trade-off between speed and performance (see the “-fast” variant).
>
> For extremely difficult cases where both the prior network and 3DGS may struggle, we see future potential in integrating sparse LiDAR measurements to build more reliable priors. We plan to explore this direction further as part of future work.
>
> ---
>
>
> >**Q2: Although the paper mentions that model-generated priors contain noise, the analysis of how different levels of prior noise affect reconstruction quality is insufficient. A more detailed ablation study on the impact of prior accuracy across various scenes would strengthen the understanding of the method’s robustness in noisy conditions.**
>
> Thanks for the feedback. Since our method uses multi-view consistency to filter out noise in the prior, it is inherently robust to prior noise. The performance differences between different priors mainly arise because more accurate priors preserve larger valid regions after the multi-view geometric check, which in turn leads to better optimization of 3DGS. We analyze how prior quality affects final performance in Appendix Table 9. Additionally, in the following table, we quantify the accuracy of the prior depths to provide a clearer understanding of our method’s robustness to prior noise. These new results will be included in the revised version as part of Appendix Table 9.
>
>
>
> **Table: Impact of different prior depths with Eve3D. Relative error rates are reported to quantify the accuracy of prior depths.**
>
> | Prior Source       | Prior Error Rate (%) | Precision ↑ | Recall ↑ | F1 Score ↑ |
> |--------------------|--------------------|-------------|----------|-------------|
> | FoundationStereo   | 5.17               | 0.553       | 0.631    | 0.581       |
> | Stereo Anywhere    | 6.81               | 0.533       | 0.600    | 0.555       |
> | MVSAnywhere        | 5.91               | 0.506       | 0.578    | 0.532       |
> | MVSFormer          | 7.61               | 0.493       | 0.598    | 0.528       |
> | OMNI-DC            | 8.36               | 0.448       | 0.540    | 0.479       |
>
>
>
> ---
>
> >**Q3: The experiments primarily focus on standard benchmarks, but the method’s performance in extreme scenarios (e.g., highly dynamic environments, low-light conditions, or extremely large-scale scenes) remains unaddressed. The paper does not provide evidence of its effectiveness beyond the tested datasets, limiting confidence in its broader applicability.**
>
> We acknowledge that dynamic environments and extremely large-scale scenes represent important challenges, but these are beyond the scope of this paper. Our focus is on evaluating performance across standard benchmarks that are commonly used to compare state-of-the-art methods (e.g., 3DGS, GOF, PGSR, Neuralangelo, etc), which also do not explicitly address these extreme scenarios. We consider the extension of our framework to dynamic or large-scale settings as an exciting direction for future work. Here we provide a preliminary experiment, based on the VastGaussian implementation for large-scale scenes. We follow VastGaussian to divide the large-scale scene into $2\times2$ blocks to train our base model and Eve3D. As shown in the table, Eve3D still improves the large-scale surface reconstruction quality.
>
>
> **Table: Eve3D on VastGaussian CUHK UPPER scene of GauU-Scene dataset**
>
> | Method                           | VastGaussian + our base model | VastGaussian + Eve3D | CityGaussianV2 |
> |----------------------------------|-------------------------------|------------------------|-----------------|
> | F1 Score ↑                   | 0.416                         | **0.519**              | 0.492           |
>
> ---
>
>
> >**Q4: How would the virtual camera affect the final results? Are the final results sensitive to the different choice of virtual cameras? An ablation study of different virtual cameras should be conducted.**
>
>
>
> In the submission, we set the baseline length as 7 \% of the scene radius for all scenes. We assess the impacts of using different baseline lengths in the following table, which illustrates that our method is robust to baseline length selections. Our method optimizes the prior depth under multi-view consistency constraints, thereby increasing the robustness to the initial priors. We will add more details of baseline length in the revised appendix.
>
>
> **Table: Impacts of baseline length**
>
> | Baseline Length         | 3% camera extent | 7% camera extent | 10% camera extent |
> |--------------------------|------------------|------------------|-------------------|
> | F1 Score ↑           | 0.580            | 0.581            | 0.581             |
>
>
> ---
>
>
> >**Q5:The generalizability of this method should be well studied, as depth estimation may fail to predict the right depth scale. Are the results sensitive to the initialization of depth estimation?**
>
>
> We study this in the appendix (Table 9), where we evaluate our method with various sources of depth priors: alternative stereo networks (Stereo Anywhere), multi-view stereo methods (MVSAnywhere and MVSFormer), and a depth completion network (OMNI-DC).
>
> It is worth noting that all of these methods predict depth maps at the correct scale, although according to different working principles: stereo models estimate disparity maps from stereo images rendered using intrinsics and extrinsics data at the same scale as the pretrained 3DGS, therefore disparity is triangulated into depth using the known focal length from 3DGS and the baseline used during rendering; multi-view stereo methods exploit the same camera poses used to optimize the 3DGS, therefore predicting depth maps at the same scale; finally, depth completion models predict dense depth maps starting from sparse measurements -- i.e., COLMAP points used to initialize 3DGS -- thus maintaining the correct scale.
>
> Despite variations in depth accuracy across these sources, our joint optimization consistently improves reconstruction performance. This demonstrates that our method is robust to different depth initializations and generalizes well beyond any specific depth estimation model. For our main results, we use FoundationStereo, which provides high-quality priors and strong generalization across datasets, as also shown in its original paper.
>
> Finally, our design allows easy integration of future, more robust depth predictors, with the potential to improve reconstruction quality further.
>
>
>
> ---
>
> >**Q6: Why did you choose depth priors other than normal or normal \& depth?**
>
> To compute the neighbor-view image correspondences for local bundle adjustment, our method requires depths as priors. In the current implementation, since we use depth priors with accurate scales, we use both depth priors and normal vectors computed from depth priors for optimization.
>
> ---

---

> > ### Comment · Reviewer_Bwjv · 2025-08-06
> >
> > Thanks for your detailed reply. Most of my concerns are addressed. I would like to keep the original score as borderline accept.

---

> ### Comment · Area_Chair_Gk5A · 2025-08-06
> **Please share your thoughts about the rebuttal and the paper**
>
> Dear Reviewer Bwjv,
>
> The author-reviewer discussion ends in two days, please read the rebuttal and share your thoughts here. Your timely response is important to the reviewing process.
>
> Thanks,
>
> AC

---

### Official Review · Reviewer_NuAQ · 2025-07-01

**Clarity:** 3
**Significance:** 3
**Originality:** 3
**Rating:** 5
**Confidence:** 4

**Summary:**

The authors propose Eve3D, a 3D reconstruction method which jointly optimizes 3DGS and the depth priors supervised by 3DGS. It renders stereo pairs with a pretrained 3DGS and acquires initial depth maps from FoundationStereo. The depth maps are subsequently refined with local bundle adjustment in high-quality regions and guided by 3DGS depth in low-quality regions to improve multi-view consistency, which then serve as depth and normal supervision for 3DGS. The authors test Eve3D on DTU, Tanks and Temples, and Mip-NeRF 360 datasets, where it outperforms its counterparts in terms of reconstruction quality on most of the scenes. Substantial ablations are performed to validate the effectiveness of joint optimization and local bundle adjustment.

**Questions:**

1. Is the local bundle adjustment applied to the learnable depth priors only, or to 3DGS depth as well? Section 3.2 only mentions the depth priors, while supp. material mentions that both are optimized (L92-93).
2. What are the specific settings for the ablations in Table 4, middle group (without the single-view prior loss)? Is the local bundle adjustment in this case only applied to 3DGS? Why would joint optimization make a difference here if the depth priors are unused in supervision?
3. As discussed above, please clarify whether the training time reported takes pretraining and depth prior computation into account. Adding also a breakdown of preprocessing time (3DGS pretraining, stereo pairs rendering, FoundationStereo prediction, triangulation) would be more helpful.
4. If possible, please add some discussions on the robustness of Eve3D in sparse-view settings (or when pretrained 3DGS/depth priors have poor quality), including visualizations and failure cases, if any.
5. What is the number of neighboring views used in local bundle adjustment for the main results reported?
6. For Fig. 8 in supp. material: it would be more informative to present a metrics vs. time plot with more data points till convergence, given that the number of neighbors used affects the training speed. Better if results of PGSR using more than one neighbor view per iteration are also presented.
7. Is the confidence mask (L183) used solely in $\mathcal{L}_{pull}$ (Eq. 12)? L280-284 hints that $\hat{\boldsymbol{M}}^c$ controls prior supervision regions, but this is not reflected in Eq. 13 and Eq. 14. Also, what is $\Gamma$ in Eq. 14?
8. L214, 216: Should be $T_{joint}$ instead of $T_{switch}$?

**Ethical Concerns:**

["NO or VERY MINOR ethics concerns only"]

**Final Justification:**

I retain my initial rating as my concerns on clarity, robustness, and computational efficiency have been addressed after reading the rebuttals and other reviews.

**Limitations:**

Yes.

**Paper Formatting Concerns:**

No major formatting issues.

**Quality:**

4

**Strengths And Weaknesses:**

Strengths:
- The idea of optimizing high-quality depth priors jointly with 3DGS to handle multi-view inconsistency and compensate for inaccuracies in the priors is straightforward but solid. The reported reconstruction performance on DTU and Tanks & Temples is also impressive, where the method achieves very competitive results even in equal-time comparisons (the Eve3D-fast variant).
- Comprehensive ablations are conducted to study the effect of different depth priors, local bundle adjustment, and joint optimization. The results, in general, demonstrate the benefits of refining depth priors with local bundle adjustment and optimizing jointly with 3DGS, compared to previous works.

Weaknesses:

I do not observe any major weakness. Some minor issues include:
- As Eve3D’s performance depends on the quality of initial depth priors to some degree (as discussed in supplementary material), which further depend on the stereo pairs rendered by pretrained 3DGS, it would be helpful to show and discuss how robustly Eve3D performs with sparse input views. In cases where the rendered stereo pairs are of low quality, it might be interesting to compare with priors from monocular depth estimation (or feeding identical views to the stereo models).
- Pretraining 3DGS to render the stereo pairs and attaining the depth maps from FoundationStereo incur extra (though potentially minor) overhead on the pipeline. Currently there is little information on the time taken in this stage, and whether it is included in the reported Eve3D training time is unclear.
- Some clarity-related issues, please refer to the questions part.

---

> ### Author Rebuttal · Authors · 2025-07-30
>
> ### **Responses to Reviewer NuAQ** ###
>
> We thank the reviewers for their insightful feedback. Below we address the questions and concerns raised.
>
> ---
>
> >**Minor issues**
>
> >- As Eve3D’s performance depends on the quality of initial depth priors to some degree (as discussed in supplementary material), which further depend on the stereo pairs rendered by pretrained 3DGS, it would be helpful to show and discuss how robustly Eve3D performs with sparse input views. In cases where the rendered stereo pairs are of low quality, it might be interesting to compare with priors from monocular depth estimation (or feeding identical views to the stereo models).
>
>
> >- Pretraining 3DGS to render the stereo pairs and attaining the depth maps from FoundationStereo incur extra (though potentially minor) overhead on the pipeline. Currently there is little information on the time taken in this stage, and whether it is included in the reported Eve3D training time is unclear.
>
> Thanks for these observations. We are going to answer both points according to the questions raised below.
>
> ---
>
> >**Q1: Is the local bundle adjustment applied to the learnable depth priors only, or to 3DGS depth as well? Section 3.2 only mentions the depth priors, while supp. material mentions that both are optimized (L92-93).**
>
>
> Both learnable depth priors and 3DGS depth are optimized by local bundle adjustment. Before $T_{joint}$, local bundle adjustment is applied only between depth priors to provide a multi-view consistent initialization (main paper, L175-179). After $T_{joint}$, local bundle adjustment is applied between depth priors and 3DGS depth to encourage multi-view consistency for both priors and 3DGS itself (main paper, L139-141). We will make the descriptions clearer in the revision.
>
> ---
>
> >**Q2: What are the specific settings for the ablations in Table 4, middle group (without the single-view prior loss)? Is the local bundle adjustment in this case only applied to 3DGS? Why would joint optimization make a difference here if the depth priors are unused in supervision?**
>
> The depth priors are used in both the Local Bundle Adjustment and Joint Optimization columns. The difference is that 'Joint Optimization' treats the prior depths as learnable parameters, thus the priors and 3DGS depths are jointly optimized. 'Local Bundle Adjustment' uses prior depths as source views and 3DGS depths as reference views, but this does not necessarily mean that the priors are treated as learnable parameters and jointly optimized. The ablation study with and without 'Joint Optimization' demonstrates the effectiveness of setting priors as learnable parameters. We will clarify these settings in the revision.
>
> ---
>
>
> >**Q3: As discussed above, please clarify whether the training time reported takes pretraining and depth prior computation into account. Adding also a breakdown of preprocessing time (3DGS pretraining, stereo pairs rendering, FoundationStereo prediction, triangulation) would be more helpful.**
>
> We confirm that the reported training time includes all steps in the pipeline: 3DGS pretraining, stereo pair rendering, FoundationStereo predictions, and the final 3DGS training. For Tanks and Temples, it takes an average of 4 minutes to pretrain 3DGS with Mini-Splatting2 and 8 minutes for stereo pair rendering and stereo depth predictions. The final training time for Eve3D-fast is 8 minutes (for a total of 20 minutes), and for Eve3D is 60 minutes (for a total of 1.2 hours). For DTU datasets, it takes an average of 3 minutes to pretrain 3DGS with Mini-Splatting2, 4 minutes for stereo pair rendering and stereo depth predictions, and the final training time takes 8 minutes (for a total of 15 minutes). We will include a detailed timing breakdown of each preprocessing step in the appendix to provide better transparency.
>
> ---
>
> >**Q4: If possible, please add some discussions on the robustness of Eve3D in sparse-view settings (or when pretrained 3DGS/depth priors have poor quality), including visualizations and failure cases, if any.**
>
>
> Although sparse-view reconstruction is not Eve3D’s primary target, we address the reviewer’s interest in robustness under challenging conditions by providing this analysis. In particular, we evaluate the robustness of Eve3D on the sparse 3-view DTU datasets. The comparison results are shown in the following table. We use the same hyperparameter setting as the dense-view DTU reconstruction. With only 3-view inputs, Eve3D still significantly improves the reconstruction quality from 1.82 mm to 1.08 mm. Eve3D also achieves comparable performances to FatesGS and Sparse2DGS (outperforming the latter), which are specifically designed for sparse-view surface reconstruction, whereas our method is not. The FoundationStereo can utilize robust monocular information from DepthAnythingV2, and we use the original RGB images as the left image of the stereo pair when performing stereo matching, which reduces the impact of poor 3DGS rendering quality on depth estimation. In the subsequent joint optimization, both prior depth and 3DGS depth are constrained by multi-view consistency. The monocular information from clean left-view images and multi-view consistency constraints improves the robustness of Eve3D to sparse-view cases.
>
>
> **Table: Reconstruction quality on the sparse 3-view DTU setting**
>
> | Methods             | our base w/o Eve3D | Eve3D | 3DGS | 2DGS | FatesGS | Sparse2DGS |
> |---------------------|--------------------|--------|------|------|---------|-------------|
> | **CD (mm) ↓**       | 1.82               | 1.08   | 3.54 | 1.69 | 0.92    | 1.13        |
>
>
> ---
>
>
> >**Q5: What is the number of neighboring views used in local bundle adjustment for the main results reported?**
>
> The main results are reported using 4 neighboring views, we will clarify this in the text.
>
> ---
>
> >**Q6: For Fig. 8 in supp. material: it would be more informative to present a metrics vs. time plot with more data points till convergence, given that the number of neighbors used affects the training speed. Better if results of PGSR using more than one neighbor view per iteration are also presented.**
>
>
> Thanks for the valuable suggestion. Unfortunately, we cannot provide a new figure during the rebuttal stage, but we will add the requested figure to the appendix in the revised version of the paper. For example, Eve3D needs 8 minutes and PGSR needs 7 minutes for training 5k iterations, and Eve3D needs 60 minutes and PGSR needs 45 minutes for training 30k iterations. For the convergence, both methods achieve the performance close to the final 30K iteration after 20k iterations. We will add more data points with running time to illustrate the convergence conditions from 10k to 30k iterations in Fig. 8.
>
>
> Our method jointly optimizes the depth priors and the 3DGS representation, where the prior depths are independent parameters. As a result, incorporating multiple neighboring views per iteration leads to more effective optimization. While PGSR imposes consistency between depths rendered from the shared 3DGS representation. Including more views in PGSR would require additional forward rendering passes per iteration, introducing extra computational overhead. Moreover, since all views share the same 3DGS geometry, the benefit of adding more views for regularization is limited compared to our approach.
>
> **Table: PGSR with more than one neighbor views**
>
> | Method              | F1 Score ↑ | Time |
> |--------------------|--------|------|
> | PGSR               | 0.516  | 45m  |
> | PGSR (4 neighbors) | 0.519  | 2h   |
>
>
>
> ---
>
> >**Q7: Is the confidence mask (L183) used solely in $\mathcal{L}_{pull}$ (Eq. 12)? L280-284 hints that $\hat{M}^C$ controls prior supervision regions, but this is not reflected in Eq. 13 and Eq. 14. Also, what is $\Gamma$ in Eq. 14?**
>
> Thank you for pointing this out. The confidence mask introduced in Line 183 is indeed also applied to Eq.13 and Eq.14, where both losses are computed over confident regions. We will revise the paper to clearly indicate that the mask is used in Eq.13 and 14. $\Gamma(A,B)$ is defined as $(1-A^TB)$ -- i.e., one minus the dot product between pixel-wise vectors in normals maps A and B.
>
> ---
>
> >**Q8: L214, 216: Should be $T_{joint}$ instead of $T_{switch}$ ?**
>
>
> This is indeed a notation error — it should be $T_{\text{joint}}$ throughout. We thank the reviewer for pointing this out and will fix this inconsistency in the revised version.
>
> ---

---

> > ### Comment · Reviewer_NuAQ · 2025-08-06
> >
> > I appreciate the thorough responses from the authors, which largely address my questions regarding clarity, computational overhead, and robustness of the method. So far, I have no further questions after reading other reviews and comments and thus incline to retaining my previous rating of acceptance.

---

> ### Comment · Area_Chair_Gk5A · 2025-08-06
> **Please share your thoughts about the rebuttal and the paper**
>
> Dear Reviewer NuAQ,
>
> The author-reviewer discussion ends in two days, please read the rebuttal and share your thoughts here. Your timely response is important to the reviewing process.
>
> Thanks,
>
> AC

---

### Official Review · Reviewer_p1BA · 2025-07-02

**Clarity:** 2
**Significance:** 2
**Originality:** 3
**Rating:** 4
**Confidence:** 4

**Summary:**

This paper proposes Eve3D, a dense 3D reconstruction approach based on 3D Gaussian Splatting. Given that the depth priors obtained from pre-trained vision models are imperfect, a joint optimization mechanism is proposed in Eve3D, where the priors first improve the quality of 3D GS, which in turn refine the priors. This iterative process gradually improves the 3D reconstruction quality. Moreover, a bundle adjustment based optimization step is also introduced to overcome the local supervision of 3DGS. Experimental results on T&T, DTU and Mip-NeRF360 datasets demonstrate the effectiveness of the proposed method.

**Questions:**

Please refer to the weaknesses.

**Ethical Concerns:**

["NO or VERY MINOR ethics concerns only"]

**Final Justification:**

Confusion over the method is resolved in the rebuttal, authors are encouraged to include the explanation to improve the presentation for final version if the paper gets accepted.

**Limitations:**

Yes

**Quality:**

3

**Strengths And Weaknesses:**

Strengths
1. Extensive experiments have been conducted on three benchmarks, and the proposed method achieves better performance compared to existing works.
2. Detailed ablation studies have been provided as well as the corresponding analysis.

Weaknesses
1. The presentation of the method is not very clear. There are different stages involved in the whole pipeline, which includes optimizing over 3D Gaussians and the learnable depth. However, it is not clear which losses are used at different stages, does the loss function in Eq.(10) only responsible for refining the learnable depth prior or both? After switching to optimizing the learnable depth prior, will it still switch back to optimize 3DGS? The optimizing targets are different before and after T_joint  to the text, but is not expressed in Eq (13)? In addition, Figure 2 is not clear about the alternative optimization.
2. The proposed method involves complicate training process with many loss terms. Are the weights for different losses set to one according to Eq.(13)?  Are these weights values applicable to different datasets or hyperparameter tuning is required? How to decide the T_joint and how does it affect?
3. The introduction of depth-normal consistency loss and the scale loss should be self-contained by either including references or descriptions.
4. Missing references.
VCR-GauS: View Consistent Depth-Normal Regularizer for Gaussian Surface Reconstruction. Neurips 2024.

---

> ### Author Rebuttal · Authors · 2025-07-30
>
> ### **Responses to Reviewer p1BA** ###
>
> We thank the reviewers for their insightful feedback. Below we address the questions and concerns raised.
>
> ---
>
> >**Q1: The presentation of the method is not very clear. There are different stages involved in the whole pipeline, which includes optimizing over 3D Gaussians and the learnable depth. However, it is not clear which losses are used at different stages, does the loss function in Eq.(10) only responsible for refining the learnable depth prior or both? After switching to optimizing the learnable depth prior, will it still switch back to optimize 3DGS? The optimizing targets are different before and after $T_{joint}$ to the text, but is not expressed in Eq (13)? In addition, Figure 2 is not clear about the alternative optimization.**
>
> We acknowledge that our description of the optimization stages could be clearer, and we will improve this presentation in the revision. To address the reviewer's specific concerns:
>
> **Regarding the loss functions at different stages:** Eq. (13) already reflects the different optimization targets. Before $T_{joint}$, rendered depth $D$ is supervised by fixed stereo depth priors $D^* $ using $||D^* - D||$, optimizing only 3DGS parameters. After $T_{joint}$, we switch to $||\hat{D} - D||$ (where $\hat{D}$ is a learnable depth map initialized with $D^*$), enabling joint optimization of both $D$ and $\hat{D}$ through backpropagation. The local bundle adjustment loss Eq. (10) is applied on prior depths to provide multi-view consistent prior initialization before $T_{joint}$, and then optimizes both rendered depths and prior depths jointly after $T_{joint}$.
>
> **Regarding the optimization schedule:** After switching to $T_{joint}$, there is no alternating optimization of depth priors and 3DGS parameters: we continue with joint optimization of both for all the remaining iterations, where both single-view prior loss Eq. (13) and local bundle adjustment loss Eq. (10) operate on both 3DGS and prior depths simultaneously after $T_{joint}$.
>
> **Regarding Figure 2 clarity:** We recognize that Figure 2 does not clearly show the temporal aspects of our method. The bidirectional arrows between "Learnable Depth Priors" and "3DGS" represent the joint optimization phase after $T_{joint}$, but we failed to clearly indicate that before $T_{joint}$, supervision occurs only in a single direction (from fixed priors to 3DGS). We will revise Figure 2 to explicitly show these two phases and add timeline indicators to eliminate any remaining ambiguity.
>
> ---
>
> >**Q2: The proposed method involves complicate training process with many loss terms. Are the weights for different losses set to one according to Eq.(13)? Are these weights values applicable to different datasets or hyperparameter tuning is required? How to decide the T_joint and how does it affect?**
>
> Thanks for pointing this out. We use the same set of weight values in all datasets without further tuning required. In our final loss, the weight of color loss $L_{c}$ is one. The depth-normal consistency loss $L_{dn}$ and scale loss $L_{s}$ are from PGSR implementations. We follow PGSR to set the loss weights $\lambda_{dn}=0.015, \lambda_{dn}=100.0$. For single-view prior loss, pull loss and local bundle adjustment loss, we set the weight values $\lambda_{prior}=0.05, \lambda_{pull}=0.05, \lambda_{lba}=0.15$. The introduction of $T_{joint}$ is due to 3D Gaussians are initialized from point cloud and need a few iterations of optimization to render reasonable contents. We set the default value of $T_{joint} = 7k$, primarily to align with the iteration at which the depth-normal consistency loss begins, following the practice of 2DGS, GOF, and PGSR. The choice of $T_{joint}$ is robust because, with local bundle adjustment providing multi-view consistency supervision, it is not necessary for 3DGS to be perfectly optimized. For example, the Eve3D-fast variant reduces the total number of iterations from 30k to 5k, sets $T_{joint} = 1k$, and still achieves good performance. We will add more details and discussions of the hyperparameter in the revised implementation details.
>
> ---
>
> >**Q3: The introduction of depth-normal consistency loss and the scale loss should be self-contained by either including references or descriptions.**
>
> Thank you for this suggestion. The depth-normal consistency is from PGSR implementation, and similar implementations are also used in 2DGS, GOF. The scale loss is from PGSR. We will add detailed descriptions and references of the depth-normal consistency loss and scale loss to make the section more self-contained.
>
> ---
>
> >**Q4: Missing references. VCR-GauS: View Consistent Depth-Normal Regularizer for Gaussian Surface Reconstruction. Neurips 2024.**
>
> Thanks for pointing this out. We will include the VCR-GauS reference in our revision as it is indeed to our depth-normal consistency formulation.
>
> ---

---

> > ### Comment · Reviewer_p1BA · 2025-08-06
> >
> > Thank the authors for the detailed responses. My main concern over the paper is the clarity of the presentation of the proposed method, and it has been addressed in the rebuttal. The authors are encouraged to include those details in the final version if the paper gets accepted.

---

### Comment · Area_Chair_Gk5A · 2025-08-05
**Please participate in discussions**

Dear Reviewers,

Thanks again for serving for NeurIPS, please read the rebuttal and discuss with the authors if you have any follow-up questions. The deadline of author-reviewer discussion is Aug 8, 11.59pm AoE.

Thanks,

AC

---

### Note · Authors · 2025-08-12

We sincerely thank the AC and reviewers for their time and efforts throughout the review process. We greatly appreciate that our responses and additional experiments have **addressed the concerns raised across all reviews**.

We are encouraged that the reviewers found our paper achieves impressive reconstruction results (p1BA,NuAQ,Bwjv,wAXX), supported by detailed and comprehensive ablation studies (p1BA,NuAQ), the idea is solid (NuAQ) and interesting (wAXX), novel apporach and addresses the limitations of existing methods (Bwjv).

---

Below, we summarize the main concerns raised in the rebuttal along with our responses.

**Clarity of the method presentation** (p1BA,NuAQ): We clarified the implementation details of our method and explained our hyperparameter selection strategy. Specifically, the same set of hyperparameters was used consistently across all datasets without requiring further per-dataset tuning.


**Impact and details of prior accuracy** (NuAQ,Bwjv,wAXX): Our method parameterizes and optimizes priors during training, combined with a multi-view geometric consistency check during optimization, enabling correction of imperfect priors. We evaluated our approach in sparse-view settings where stereo depth priors are noisy and our method significantly improves surface reconstruction quality over the base model. To better illustrate the impact of prior accuracy, we provide a quantitative analysis of prior quality alongside our results on standard benchmarks. Additionally, we include a comparison between our method and directly using priors for surface reconstruction, demonstrating the necessity of incorporating priors within the joint optimization framework with 3DGS.





**Performances in extreme scenarios** (Bwjv): We conducted the experiements on CUHK UPPER scene to reconstruct large-scale scenes and our method improves the large-scale surface reconstruction quality.


**Our method with other baseline model** (wAXX): We implemented our method with 2DGS. The performance of 2DGS was significantly improved by our joint optimization method, further demonstrating the generalizability of the proposed approach.

---

Thanks again for the valuable feedback and constructive suggestions provided throughout the reviews. We will update the main paper and appendix to include all the experiments and details mentioned during the rebuttal process.

---

### Decision · Program_Chairs · 2025-09-17

**Decision:**

Accept (poster)

**Comment:**

This paper proposes a dense 3D reconstruction method leveraging 3D Gaussians. The key idea is to jointly optimize imperfect depth and normal maps predicted by existing models together with 3D Gaussians. In addition, the authors introduce a local bundle adjustment strategy designed to avoid local supervision. The method is thoroughly evaluated and compared against strong baselines on standard public datasets.

During the review process, reviewers raised concerns regarding the clarity of the writing, system complexity, generalization ability, and several technical details. Most of these concerns were adequately addressed during the rebuttal, leading all reviewers to recommend acceptance.

After carefully considering the paper, the reviews, and the authors’ clarifications in the rebuttal, the Area Chair concurs that the paper makes a sufficient contribution of interest to the community. Therefore, I recommend acceptance. The authors are strongly encouraged to integrate the clarifications, additional discussions, and further results from the rebuttal into the final version to improve accessibility and impact.